# One-Shot Weighted Ensemble Estimation for Federated Quantile Regression: Optimal Statistical Guarantees under Heterogeneous Structured Data

## Abstract

Federated Quantile Regression (FQR) has emerged as a powerful modelling paradigm for estimating conditional quantiles, offering a more comprehensive understanding of response distributions than standard conditional mean regression. However, achieving communication efficiency and optimal statistical guarantees for FQR remains challenging, particularly due to the nonsmooth nature of quantile loss functions and the presence of heterogeneously structured data, where each local agent trains its conditional quantile models with distinct sets of features. In this paper, we propose a data-driven, one-shot weighted ensemble estimator for FQR that incorporates scalable weighting schemes to effectively leverage the partially observed features at each local agent, thereby enjoying both communication efficiency and estimation optimality. Theoretically, we present a unified analysis of the proposed learning procedure, establishing that the resulting estimator exhibits asymptotic normality and attains uniformly minimum variance. Furthermore, we investigate the estimator's sensitivity to perturbations introduced by local agents and derive conditions under which the estimator achieves stability and enjoys strong out-of-sample generalization. Extensive simulations and real data analysis under various scenarios validate the asymptotic normality of our estimator and demonstrate its superior estimation accuracy and uniform convergence compared to several baseline methods across a range of quantile levels.

## 1 Introduction

Federated Learning (FL) is a powerful machine learning paradigm that aims to learn a consensus model while keeping data distributed across multiple agents. The model is trained without transmitting local data over the network, thereby preserving privacy while leveraging information from participating agents to enhance estimation accuracy (Fraboni et al., 2023). Classical approaches to FL typically focus on modelling the conditional mean of the response given covariates of interest under the assumption of homogeneous covariate effects. However, the assumption of homogeneous covariate effects is often not applicable in settings where the relationship between the response and covariates is inherently heterogeneous: covariate effects may vary significantly across different quantile levels (Wang et al., 2012; He et al., 2023). Moreover, in many scientific applications (e.g., hydrological (Weerts et al., 2011), sociological (Yang et al., 2012), and medical (Huang et al., 2017)), when the goal is to explain the extreme behaviour of a particular variable, the lower and upper quantiles of the conditional response distribution are often of greater interest than the mean, as they yield more succinct and interpretable conclusions. To capture heterogeneous covariate effects, Quantile Regression (QR) has been developed as a powerful alternative for estimating conditional quantiles of the response. In addition to capturing heterogeneity, QR provides a robustness guarantee to outliers and remains effective under skewed or heavy-tailed response distributions without requiring correct specification of the likelihood function (Koenker, 2005). These advantages make QR highly compatible with FL, where data typically originates from diverse, distributed sources, which leads to a modelling paradigm of Federated Quantile Regression (Huang et al., 2020; Shi et al., 2025; Shen et al., 2023; Tan et al., 2022).

Despite the promising theoretical and practical performance of FQR, the existing literature for FQR has largely focused on consistency, communication complexity, and algorithmic development (Shi et al., 2025; Wang & Lian, 2023; Mirzaeifard et al., 2025; Wang & Lian, 2020; Wang et al., 2021). While these works provide optimal point estimators, they fall short in quantifying uncertainty or addressing the practical challenge of inference on the effects of covariates in the conditional quantile function. We argue that a statistical guarantee is essential in FQR, particularly given that the training samples could be collected from diverse sources in FL (Ghosh et al., 2019; Tan et al., 2022). This is mainly because of its critical importance in measuring the uncertainty associated with the estimate in applications, as opposed to relying on a single-point estimate. Insight into the asymptotic distribution of the estimates provides a foundation for making more informed decisions by quantifying the uncertainty of the estimate. Meanwhile, implementing an estimate without verifying its sensitivity to perturbations can be risky. In many real-world operational settings, estimates must be carefully evaluated before deployment. Therefore, the focus is not only on obtaining optimal estimates but, more importantly, on assessing their associated statistical stability and generalization. This motivates the primary research objective of the paper to investigate the statistical guarantee of the FQR estimates.

## 1.1 Main Contribution

In this paper, we investigate statistical guarantees, particularly asymptotic distribution, stability and out-of-sample performance for FQR estimates, focusing on heterogeneously structured data environments in which local agents train QR models using distinct subsets of features. Such heterogeneity arises from both practical constraints and task-specific considerations. In the former case, agents may perform local model selection to enhance predictive performance (Wang et al., 2024). In the latter, limitations related to feasibility, privacy-preserving requirements, and resource constraints restrict the set of accessible covariates for each agent (Cheng et al., 2023) (We refer the reader to related work for further details.). To the best of our knowledge, this is the first work to consider FQR in this setting. We emphasize that this heterogeneous structure poses significant challenges, representing a marked departure from the standard FL setting, where all agents operate on an identical feature set. To address these challenges, we propose a data-driven, one-shot weighted ensemble estimator for FQR, which incorporates scalable weighting schemes to effectively leverage the partially observed feature sets across agents. We establish theoretical properties where the proposed estimator enjoys strong statistical guarantees and demonstrate its empirical effectiveness through comprehensive numerical experiments across a range of settings. Our main contributions are summarized as follows.

1. We propose a communication-efficient weighted ensemble estimator for federated QR, designed for heterogeneous data environments where local agents train QR models on distinct feature subsets.

2. Theoretically, we do a rigorous analysis of the proposed method, showing that the resulting estimator exhibits asymptotic normality under any weighting scheme and attains uniformly minimum variance with the proposed optimal weighting. We further develop a foundational stability concept to assess the estimator's sensitivity to perturbations from local agents and establish that the proposed estimator achieves stability and enjoys strong out-of-sample generalization.

3. Numerical experiments demonstrate that the proposed weighted ensemble estimator outperforms several baseline methods in estimation accuracy and uniform convergence across various quantile levels.

## 1.2 Related Work

This paper is motivated by the significance of QR in federated learning applications and the practical need to handle heterogeneous, structured data settings for distributed estimation and inference. In this section, we review lines of work most closely related to this paper.

**Statistical inference for FQR.** Statistical inference for FQR is widely recognized as an important yet challenging task. This challenge arises from the decentralized feature of data in FL (McMahan et al., 2016), rendering existing methodologies inapplicable. Some algorithms have been proposed

to be compatible with distributed architectures (Jordan et al., 2019; Fan et al., 2023), but they are not applicable to FQR due to their requirements on the loss function, typically assuming strong convexity and twice differentiability with Lipschitz-continuous second derivatives. To address the challenges posed by the nonsmooth loss function, one line of research focuses on a smoothing technique to make the loss function convex and differentiable. Specifically, Tan et al. (2022) leverages a double-smoothing approach to achieve optimal inference in distributed quantile regression. However, such a technique could cause smoothing bias, primarily affecting the estimation, especially in the heterogeneous structured data setting (Fernandes et al., 2021; He et al., 2023). An alternative approach employs meta-analysis techniques that average estimates from separate data sources to obtain synthesized estimators of QR coefficients. Although it offers the advantage of communication efficiency, it requires stringent scaling to achieve the desired theoretical guarantees. Furthermore, Jordan et al. (2019) highlighted that a stringent constraint on the number of sources is imposed to ensure the optimal convergence rate: the number of agents is assumed to be far fewer than the total sample size. This paper addresses the limitations of smoothing techniques and the stringent constraints in the context of FQR, enabling distributed estimation with optimal statistical guarantees. The core innovation of the proposed approach lies in estimating the FQR coefficients, using a one-shot weighted ensemble method that leverages the information of observed features at each local agent. Notably, the proposed estimator relaxes the stringent constraint on the number of agents while preserving communication efficiency, requiring only a single round of communication.

**Heterogeneous structured data.** The heterogeneous structured data we investigate is motivated by practical constraints and a series of studies addressing similar data across a broad range of applications without necessarily being referred to by this name, including decentralized clinical trials (DCT) (De Jong et al., 2022), structured missing data (Cheng et al., 2023), model aggregation (Le & Clarke, 2022; Ding et al., 2022), and selective inference (Wang et al., 2024). Specifically, motivated by the need to adjust the model selection process, Wang et al. (2024) developed a selective inference tool to infer the effects of selected variables on conditional quantile functions, aiming to ensure reliable inference post-selection. For model aggregation, Ding et al. (2022) introduced the concept of 'multiviews' and proposed a new method for supervised learning with multiple sets of features, which is particularly important in biology and medicine, where experts from different backgrounds have their perspectives on the selection of variables. However, a major difference between this line of work and ours is that most estimators are trained using the same set of observations, while ours is trained on each agent's own data, with only final outputs shared. We emphasize that the decentralized nature of the data in this paper presents additional challenges in theoretical and methodological development, particularly in quantifying correlations and developing a feasible estimator that accommodates this decentralization, such as determining and obtaining the necessary statistics for aggregating the final output. A complementary work by (Cheng et al., 2023) proposed a method for collaboratively learning least squares estimates for agents, where each agent observes a different subset of features due to missingness. While similar in setting, we develop an estimator that considers broader practical constraints and task-specific considerations, making our approach adaptive and scalable.

## 2 PRELIMINARIES

In this section, we introduce the preliminaries and notation that will be used throughout the paper.

**Quantile Regression.** Let $x \in \mathbb{R}^d$ be a $d$-dimensional covariate vector and $y \in \mathbb{R}$ a scalar response variable. We aim to estimate the $\tau$-th conditional quantile of $y$ given $x$ at a pre-specified quantile level $\tau \in (0, 1)$, focusing on the linear QR model of $Q_\tau(y \mid x) = x^\top \beta^\star(\tau)$, where $\beta^\star(\tau) = (\beta_1^\star(\tau), \ldots, \beta_d^\star(\tau))^\top \in \mathbb{R}^d$ is a vector of unknown parameters. This model can be equivalently expressed as:

$$y = x^\top \beta^\star(\tau) + \xi(\tau), \tag{1}$$

where $\xi(\tau) \in \mathbb{R}$ is a random error satisfying $\mathbb{P}\{\xi(\tau) \leq 0 \mid x\} = \tau$ (Koenker, 2005). In other words, the conditional $\tau$th quantile of each $\xi(\tau)$ given $x$ is zero. The special case $\tau = 1/2$ corresponds to median regression. Let $\rho_\tau(u) = u\{\tau - I(u < 0)\}$ denote the non-differentiable check loss function, where $I(\cdot)$ denotes the usual indicator function. Given the distribution function of $y$, $\beta^\star(\tau)$ can be obtained by solving

$$\beta^\star(\tau) = \underset{\beta \in \mathbb{R}^d}{\arg\min} \mathbb{E}\left[\rho_\tau(y - x^\top \beta(\tau))\right].$$

Suppose we consider $M$ agents, each with an identical sample size $n$ for simplicity. Let $\{(x_{i,m}, y_{i,m})\}_{i=1}^n$ denote $n$ independent and identically distributed (i.i.d) samples from agent $m$, $\forall m \in [1, M]$. Define $N = nM$, $Y_m = (y_{1,m}, \ldots, y_{n,m})^\top \in \mathbb{R}^n$, $X_m = (x_{1,m}, \ldots, x_{n,m})^\top \in \mathbb{R}^{n \times d}$.

**Heterogeneous Structured Data.** We consider an FL problem with heterogeneous structured data, where each agent observes or selects only a subset of the full feature due to data collection constraints or selective biases. Each agent's data is assumed to follow the linear QR model equation 1. To mathematically formalize this feature-wise data partitioning and operationalize the ideas of Cheng et al. (2023), we introduce a permutation matrix $\Pi_m \in \mathbb{R}^{d \times d}$ for each agent $m \in \{1, \cdots, M\}$. Specifically, define

$$\Pi_m^\top := \begin{bmatrix} \Pi_{m+}^\top & \Pi_{m-}^\top \end{bmatrix}, \quad \Pi_{m+} \in \mathbb{R}^{d_m \times d}, \quad \Pi_{m-} \in \mathbb{R}^{(d-d_m) \times d},$$

where $\Pi_{m+}$ extracts the observed features (covariates) and $\Pi_{m-}$ the unobserved ones for agent $m$. Let $\Sigma$ be the covarince matrix of $x_{i,m}$, i.e., $\mathbb{E}(x_{i,m} x_{i,m}^\top) = \Sigma$. Given a sample $(x_{i,m}, y_{i,m}) \in \mathbb{R}^d \times \mathbb{R}$, the covariate vector is decomposed as

$$x_{i,m} = \Pi_m^\top \begin{bmatrix} x_{i,m+} \\ x_{i,m-} \end{bmatrix},$$

where $x_{i,m+} = \Pi_{m+} x_{i,m} \in \mathbb{R}^{d_m}$ and $x_{i,m-} = \Pi_{m-} x_{i,m} \in \mathbb{R}^{d-d_m}$ represent the observed and unobserved features, respectively, along with the associated response $y_{i,m}$ and corresponding marginal covariance

$$\Sigma_{m+} := \mathbb{E}\left[x_{i,m+} x_{i,m+}^\top\right] = \Pi_{m+} \Sigma \Pi_{m+}^\top,$$

which can be estimated from local data. We emphasize that this decomposition plays a central role in the design of the learning algorithms proposed in later sections, which rely solely on observed covariates while preserving the global inference objective.

For notational simplicity, for any vector $v \in \mathbb{R}^d$, we define the projections $v_{m+} := \Pi_{m+} v$ and $v_{m-} := \Pi_{m-} v$. These definitions extend analogously to matrix-valued notation, and we further define that, for any matrix $A \in \mathbb{R}^{d \times d}$, suppose

$$A_{m+} := \Pi_{m+} A \Pi_{m+}^\top, \quad A_{m-} := \Pi_{m-} A \Pi_{m-}^\top,$$
$$A_{m\pm} := \Pi_{m+} A \Pi_{m-}^\top, \quad A_{m\mp} := \Pi_{m-} A \Pi_{m+}^\top.$$

For a positive semi-definite matrix $A$, we define the $A$-norm of a vector $z \in \mathbb{R}^d$ as $\|z\|_A := \sqrt{\langle z, Az \rangle}$. In addition, for any two positive semi-definite matrices $A$ and $B$, we write $A \succeq B$ to denote that $A - B$ is positive semi-definite. Table 1 summarizes the notations adopted throughout the paper.

Table 1: Notations and their meaning

| Notations | Meaning |
| --- | --- |
| $\tau$ | quantile level |
| $y_{i,m}$ | $i$-th observed response for agent $m$ |
| $x_{i,m+}, x_{i,m-}$ | local observed and unobserved features for agent $m$ |
| $\beta^\star(\tau)$ | true parameters |
| $\tilde{\beta}_m(\tau)$ | local QR estimator for agent $m$ |
| $\widehat{\beta}(\tau; \Omega(W))$ | global estimator |
| $N, n$ | total and local sample size |
| $M$ | number of agents |
| $\Sigma_{m+}, \Sigma_{m-}$ | observed and unobserved covariance for agent $m$ |
| $\Pi_m$ | permutation matrix for agent $m$ |
| $\Pi_{m+}, \Pi_{m-}$ | extract observed and unobserved features (covariates) for agent $m$ |

## 3 METHODOLOGY

The key challenge in designing an estimator for our setting lies in effectively integrating partially observed feature information to ensure statistical optimality, while maintaining high communication efficiency. On the communication side, efficiency becomes particularly critical in large-scale networks with numerous local data-collecting entities, especially under bandwidth constraints. To enable scalability, it is important to minimize the number of communication rounds and offload computationally intensive tasks to local machines without compromising statistical accuracy. Regarding statistical optimality, we argue that a desirable method should not only ensure asymptotic consistency with respect to the ground truth, but more importantly, minimize the prediction error on any test sample with partial features $x_+ = \Pi_{m+}x$ observed by agent $m$. To address the aforementioned challenges, we propose a data-driven one-shot estimation procedure consisting of three steps.

**Step 1: Local estimation.** Each local agent learns its own estimate based on the subset of features it observes or selects. Correspondingly, the local QR estimator at agent $m$ is defined as

$$\tilde{\beta}_m(\tau) = \arg\min_{\beta(\tau)} \left\{ \frac{1}{n} \sum_{i=1}^{n} \rho_\tau \left( y_{i,m} - x_{i,m+}^\top \beta(\tau) \right) \right\}, \tag{2}$$

where $x_{i,m+} = \Pi_{m+}x_{i,m} \in \mathbb{R}^{d_m}$ denotes the observed feature vector for the $m$th agent.

**Step 2: Weighted ensemble estimation.** Each agent then transmits its estimate $\tilde{\beta}_m(\tau)$ to a central server. The central server aggregates the collection of local estimates $\{\tilde{\beta}_m(\tau)\}_{m=1}^{M}$ by solving a weighted optimization problem that accounts for the heterogeneous structured data across agents (see Section 3.1). This aggregation yields a global estimator $\widehat{\beta}(\tau)$, which integrates information from all agents while respecting their partial feature access.

**Step 3: Model distribution.** Finally, the central server distributes the global estimator $\widehat{\beta}(\tau)$ and its appropriately transformed versions $T_m\widehat{\beta}(\tau)$ to each agent. The specific form of the transformation operator $T_m$ is provided in Section 3.1. These agent-specific transformations enable each node to make predictions using only its locally observed features, while still benefiting from the information encoded in the full feature space.

**Remark 3.1.** *We emphasize that the communication cost depends solely on the local dimension $d_i$ and does not scale with $n$, $d$, or $m$, thereby ensuring efficiency. Specifically, in the first communication round, agent $i$ transmits $d_i^2 + 2d_i$ scalars to the central server, and in the second round, the server returns the updated local parameter vector of size $d_i$. Consequently, the total per-agent communication cost is $d_i^2 + 3d_i$.*

### 3.1 WEIGHTED ENSEMBLE ESTIMATION

**Prediction error.** The primary objective is to design an estimator, $\widehat{\beta}(\tau)$, that utilizes partially observed data to minimize the full-feature prediction error on a fresh sample $x_i \in \mathbb{R}^d$:

$$\mathbb{E}\left[ \left( \langle x_i, \widehat{\beta}(\tau) \rangle - \langle x_i, \beta^\star(\tau) \rangle \right)^2 \right] = \|\widehat{\beta}(\tau) - \beta^\star(\tau)\|_\Sigma.$$

We are also interested in obtaining an estimator, $\widehat{\beta}_m(\tau)$, which minimize the partial-feature prediction error on a fresh sample $x_{i,m+} = \Pi_{m+}x_{i,m}$ for agent $m$:

$$\mathbb{E}\left[ \left( \langle x_{i,m+}, \widehat{\beta}_m(\tau) \rangle - \langle x_{i,m}, \beta^\star(\tau) \rangle \right)^2 \right] = \left\| \widehat{\beta}_m(\tau) - T_m\beta^\star(\tau) \right\|_{\Sigma_{m+}}^2 + \left\| \beta_{m-}^\star(\tau) \right\|_{\Gamma_{m-}}^2,$$

where the second term $\|\beta_{m-}^\star(\tau)\|_{\Gamma_{m-}}^2$ represents the irreducible error due to unobserved features. Here, $\Gamma_{m-} := \Sigma_{m-} - \Sigma_{m\mp}\Sigma_{m+}^{-1}\Sigma_{m\pm}$ is the Schur complement, and $T_m$ is a linear transformation matrix defined as

$$T_m := \begin{bmatrix} I_{d_m} & A^{-1}B \end{bmatrix} \Pi_m,$$

where $A$ and $B$ are the weighted Hessian and covariance matrix defined as follows,

$$A := \mathbb{E}\left[ f_{\xi_{i,m}}(0|x_{i,m+})x_{i,m+}x_{i,m+}^\top \right], \quad B := \mathbb{E}\left[ f_{\xi_{i,m}}(0|x_{i,m+})x_{i,m+}x_{i,m-}^\top \right].$$

We emphasize that the operator $T_m$ plays a pivotal role in the estimation process. Specifically, $T_m\beta^\star(\tau)$ provides the best possible predictor for agent $m$ compared with the naive approach of using the subvector, $\Pi_{m+}\beta^\star(\tau)$, which simply selects the coefficients corresponding to the observed features. In contrast, $T_m\beta^\star(\tau)$ accounts for the correlations among all features, thereby improving prediction accuracy. The scalar term $f_{\xi_{i,m}}(0 \mid x_{i,m+})$ denotes the conditional density of the error $\xi_{i,m}(\tau)$ at zero, given the observed feature vector $x_{i,m+}$, and reflects the local concentration of noise around the $\tau$th quantile.

**Estimates aggregation.** We now present a weighted empirical risk minimization problem that is used to aggregate the local estimates to obtain a global estimator. Let $W_m \in \mathbb{R}^{d_m \times d_m}$ be a symmetric, positive definite weight matrix for agent $m = 1, \ldots, M$, and denote the collection of weight matrices by $\Omega(W) := \{W_m\}_{m=1}^M$. The global estimator $\widehat{\beta}(\tau) := \widehat{\beta}(\tau; \Omega(W))$ is obtained by solving the following optimization problem:

$$\widehat{\beta}(\tau; \Omega(W)) =: \arg\min_{\beta(\tau)} \sum_{m=1}^M \left\| \beta_{m+}(\tau) + (A^{-1}B)\beta_{m-}(\tau) - \tilde{\beta}_m(\tau) \right\|_{W_m}^2. \tag{3}$$

A local estimator for agent $m$ is then defined as $\widehat{\beta}_m(\tau) := T_m\widehat{\beta}(\tau; \Omega(W))$. Applying the first-order optimality condition, $\widehat{\beta}(\tau)$ admits the following closed-form expression:

$$\widehat{\beta}(\tau; \Omega(W)) = \left( \sum_{m=1}^M T_m^\top W_m T_m \right)^{-1} \left( \sum_{m=1}^M T_m^\top W_m \tilde{\beta}_m(\tau) \right). \tag{4}$$

It can be shown that $\widehat{\beta}(\tau; \Omega(W))$ is a consistent and asymptotically unbiased estimator of the true parameter $\beta^\star(\tau)$, regardless of the specific choice of weight matrices (see Lemma 4.4). Furthermore, we will show the existence of an optimal weight matrix $W^\star$ such that the corresponding estimator $\widehat{\beta}(\tau; \Omega(W^\star))$ achieves the minimum asymptotic variance among all estimators of the form $\widehat{\beta}(\tau; \Omega(W))$. The detailed procedure is summarized in Algorithm 1.

---

**Algorithm 1** One-shot Weighted Ensemble Estimation with Uniformly Minimum Variance

---

1: **Input:** Given $M$ agents, each possessing a local training dataset $\{(x_{i,m+}, y_{i,m})\}_{i=1}^n$
2: **for** $m$ in $1, \cdots, M$ **do**
3:    Compute $\tilde{\beta}_m(\tau) = \arg\min_{\beta(\tau)\in\mathbb{R}^d} \frac{1}{n}\sum_{i=1}^n \rho_\tau\left(y_{i,m} - x_{i,m+}^\top\beta(\tau)\right)$
4:    Compute $\widehat{V}_m = \frac{1}{n}\sum_{i=1}^n x_{i,m+}\left(\tau - I[y_{i,m} - x_{i,m+}^\top\tilde{\beta}_m(\tau) < 0]\right)$
5:    Compute $\widehat{R}_m = \frac{1}{n}\sum_{i=1}^n \left[f_{\xi_{i,m}}(0|x_{i,m+})x_{i,m+}x_{i,m+}^\top\right]$
6:    Transmit $\tilde{\beta}_m(\tau), \widehat{V}_m, \widehat{R}_m$ to coordinating server
7: **end for**
8: Central server constructs $\widehat{W}_m = \widehat{R}_m\left(\widehat{V}_m\widehat{V}_m^\top\right)^{-1}\widehat{R}_m$ for $m = 1, \ldots, M$
9: Central server obtain a global estimator $\widehat{\beta}^{\mathrm{OSW}}(\tau)$ through formula equation 4, and each local agent output $\widehat{\beta}_m^{\mathrm{OSW}}(\tau) = T_m\widehat{\beta}^{\mathrm{OSW}}(\tau)$

---

Compared with Cheng et al. (2023)'s work, which considers a quadratic loss for each local agent, the quantile loss used in our framework introduces substantial computational challenges in Algorithm 1. Because no closed-form expression exists for the local quantile estimator, Step 3 requires solving a linear programming problem, whereas Cheng's estimator can be computed directly via a closed-form solution. Furthermore, obtaining an estimate of the optimal weight matrix $W_m^\star$ in Step 8 requires estimating the conditional density $f_{\xi_{i,m}}(0 \mid x_{i,m+})$, a step unnecessary in Cheng's framework.

## 4 THEORETICAL PROPERTIES

In this section, we first establish the asymptotic normality of the proposed estimator $\widehat{\beta}(\tau; \Omega(W))$ and derive the optimal weight matrix that minimizes its asymptotic variance. We then introduce a consistent estimator for this optimal weight matrix to enable practical implementation. Finally, we analyze the generalization performance of the proposed estimator based on the notion of stability.

**Assumption 4.1** (Feature assumption). $x_{i,m} \sim \mathcal{N}(0, \Sigma)$, *for $i = 1, \ldots, n; m = 1, \ldots, M$.*

**Assumption 4.2** (Structural coverage). *The collection of all $M$ agents jointly spans the entire feature space.*

**Assumption 4.3** (Well-definedness). *Let $f(. \mid x_{i,m+})$ be the conditional density function of the noise $\xi_{i,m}$ given $x_{i,m+}$. Assume that this density function is continuous at 0 and $f_{\xi_{i,m}}(0 \mid x_{i,m+}) \geq \underline{f} \geq 0$ for some constant $\underline{f}$.*

Assumptions 4.1-4.3 are widely acknowledged as a regularity condition in the literature (Cheng et al., 2023; Wu et al., 2020; Xie et al., 2024). In particular, Assumption 4.1, which is also required for the least square estimation in the same settings (Cheng et al., 2023), is mild in the federated learning literature for enabling valid statistical inference. The structural assumption 4.2 ensures that the full covariance matrix $\Sigma$ can be recovered from the collection $\{\Sigma_{m+}\}_{m=1}^{M}$. Assumption 4.3 imposes the conditions that are critical for ensuring the existence of a well-defined asymptotic variance.

**Lemma 4.4** (Asymptotic consistency). *Suppose Assumptions 4.1, 4.2, and 4.3 hold. Then for any collection of positive definite weighting matrices $\Omega(W) := \{W_m\}_{m=1}^{M}$, where each $W_m \in \mathbb{R}^{d_m \times d_m}$ for $m = 1, \ldots, M$, the aggregated estimator $\widehat{\beta}(\tau; \Omega(W))$, defined in Eq. (3), is asymptotically consistent. That is, $\widehat{\beta}(\tau; \Omega(W)) \xrightarrow{p} \beta^{\star}(\tau)$.*

**Theorem 4.5** (Asymptotic normality). *Under Assumptions 4.1, 4.2, and 4.3, the aggregated estimator $\widehat{\beta}(\tau; \Omega(W))$ is asymptotically normal:*

$$\sqrt{n}\left(\widehat{\beta}(\tau; \Omega(W)) - \beta^{\star}(\tau)\right) \xrightarrow{d} N(0, C(\Omega(W))),$$

*where the asymptotic covariance matrix is given by*

$$C(\Omega(W)) = \left(\sum_{m=1}^{M} T_m^{\top} W_m T_m\right)^{-1} \left(\sum_{m=1}^{M} T_m^{\top} W_m W_m^{\star-1} W_m T_m\right) \left(\sum_{m=1}^{M} T_m^{\top} W_m T_m\right)^{-1},$$

*and*

$$W_m^{\star} = \mathbb{E}[f_{\xi_{i,m}}(0|x_{i,m+})x_{i,m+}x_{i,m+}^{\top}] \cdot V_m^{-1} \cdot \mathbb{E}[f_{\xi_{i,m}}(0|x_{i,m+})x_{i,m+}x_{i,m+}^{\top}], \tag{5}$$

$$V_m = \mathbb{E}[(x_{i,m+}(\tau - I[y_{i,m} - x_{i,m+}^{\top} T_m \beta^{\star}(\tau) < 0]))(x_{i,m+}^{\top}(\tau - I[y_{i,m} - x_{i,m+}^{\top} T_m \beta^{\star}(\tau) < 0]))]. \tag{6}$$

*Moreover, for any positive definite weight matrices $\Omega(W)$, the asymptotic covariance satisfies*

$$C(\Omega(W)) \succeq C(\Omega(W^{\star})) := \left(\sum_{m=1}^{M} T_m^{\top} W_m^{\star} T_m\right)^{-1}. \tag{7}$$

We highlight that the asymptotic normality result established in Theorem 4.5 holds for *any weighting matrices* $\Omega(W)$, underscoring the scalability of the proposed estimator. Moreover, we identify a specific optimal $W^{\star}$ as in equation 5 that minimizes the asymptotic covariance, thereby enhancing the efficiency of the estimator. Notably, Theorem 4.5 holds without stringent conditions typically required in meta-analysis (Jordan et al., 2019), which often limit agent number $M$ to be much smaller than $\sqrt{N}$.

**Remark 4.6.** *We emphasize that Theorem 4.5 is derived under the Gaussian assumption. While there exists potential to extend the proposed estimator to non-Gaussian settings, the main challenge in applying Theorem 4.5 lies in constructing the optimal weights, which in their explicit form depend on the unknown density function and parameter $\beta^{\star}$. Our analysis currently focuses on exploiting Gaussianity to render these weights estimable and thereby enable the construction of the estimator. Nevertheless, as we demonstrate in the Appendix A.5, the optimality of our approach extends beyond the Gaussian framework, highlighting the broader applicability of the proposed methodology.*

### 4.1 UNIFORMLY MINIMUM VARIANCE WEIGHTED ENSEMBLE ESTIMATION

In this section, we propose a consistent estimator of $\{W_m^{\star}\}_{m=1}^{M}$ for practical implementation.

**Lemma 4.7** (Consistent estimator). *Under the assumptions as Theorem 4.5, define*

$$\widehat{V}_m = \frac{1}{n}\sum_{i=1}^{n} x_{i,m+}\left(\tau - I\left[y_{i,m} - x_{i,m+}^{\top}\tilde{\beta}_m(\tau) < 0\right]\right), \widehat{R}_m = \frac{1}{n}\sum_{i=1}^{n} f_{\xi_{i,m}}(0|x_{i,m+})x_{i,m+}x_{i,m+}^{\top}.$$

*we have that $\widehat{W}_m := \widehat{R}_m\left(\widehat{V}_m\widehat{V}_m^{\top}\right)^{-1}\widehat{R}_m$ is a consistent estimator of $W_m^{\star}$.*

With these consistent estimators $\{\widehat{W}_m\}_{m=1}^{M}$, we define the one-shot weighted ensemble estimator with minimum asymptotic variance (OSW) for the global and local estimators as

$$\widehat{\beta}^{\text{OSW}}(\tau) := \widehat{\beta}\left(\tau; \Omega\left(\widehat{W}\right)\right), \quad \widehat{\beta}_m^{\text{OSW}}(\tau) := T_m\widehat{\beta}\left(\tau; \Omega\left(\widehat{W}\right)\right). \tag{8}$$

**Theorem 4.8** (Uniformly minimum variance estimator). *Under Assumptions 4.1, 4.2, and 4.3, the global OSW estimator $\widehat{\beta}^{\text{OSW}}(\tau)$ and local OSW estimator $\widehat{\beta}_m^{\text{OSW}}(\tau)$ are asymptotically normal:*

$$\sqrt{n}\left(\widehat{\beta}^{\text{OSW}}(\tau) - \beta^{\star}(\tau)\right) \xrightarrow{d} \mathcal{N}(0, C(\Omega(W^{\star}))),$$

$$\sqrt{n}\left(\widehat{\beta}_m^{\text{OSW}}(\tau) - T_m\beta^{\star}(\tau)\right) \xrightarrow{d} \mathcal{N}\left(0, T_m C(\Omega(W^{\star}))T_m^{\top}\right).$$

**Corollary 4.9.** *Under Assumptions 4.1, 4.2, and 4.3, the OSW estimator satisfies:*

$$\left\|\widehat{\beta}^{\text{OSW}}(\tau) - \beta^{\star}(\tau)\right\|_2 = O_p\left(\frac{1}{\sqrt{n}}\right).$$

Note that, for any $m$th agent, the local estimator $\tilde{\beta}_m(\tau)$, defined in equation 2, satisfies

$$\sqrt{n}\left(\tilde{\beta}_m(\tau) - T_m\beta_m^{\star}(\tau)\right) \xrightarrow{d} \mathcal{N}\left(0, W_m^{-1}\right).$$

As $W_m^{-1} \succeq T_m C(\Omega(W^{\star}))T_m^{\top}$, the OSW local estimator has smaller asymptotic variance. Moreover, $\widehat{\beta}_m^{\text{OSW}}(\tau)$ leverages the heterogeneous structure of each agent, thereby improving partial-feature prediction accuracy. This also highlights the benefit of tailoring the global estimator via the transformation $T_m$ for localized inference. The proposed OSW global estimator $\widehat{\beta}^{\text{OSW}}(\tau)$ reduces the overall prediction error across all features, while achieving the optimal estimation error convergence rate (Salehkaleybar et al., 2021).

## 4.2 GENERALIZATION VIA AGENT-DEPENDENT STABILITY

In this section, we establish a generalization bound for the proposed estimator based on the notion of algorithmic stability. Stability-based analyses are commonly used in statistical learning theory to derive upper bounds on generalization error, thereby ensuring out-of-sample performance. In classical settings, stability is typically defined with respect to perturbations in individual data points. However, this notion of stability does not directly apply in the FL setting, where each model is trained on agent-specific local data. To address this challenge, we define an agent-dependent stability notion tailored to FL by quantifying the effect of removing an entire agent's data. This adapts the sample-dependent stability concept from Bousquet & Elisseeff (2002); Wu et al. (2020) to our federated framework.

**Definition 4.10** (Agent-dependent stability). *Let $Z_m$ denote the dataset held by agent $m$, and $Z := \{Z_1, \ldots, Z_M\}$ the collection of all agent datasets. An FL algorithm $\mathcal{A}$ is said to be agent-dependent $\mu$-stable with respect to a loss function $\ell(\cdot)$, if for all $m = 1, \ldots, M$ and any data point $z$,*

$$\mathbb{E}_{Z,z}\left|\ell(\mathcal{A}_Z, z) - \ell(\mathcal{A}_{Z^{\backslash m}}, z)\right| \leq \mu,$$

*where $Z^{\backslash m}$ denotes the training dataset with data from agent $m$ removed and redistributed to the remaining $M - 1$ agents with the same missing structure.*

This definition captures the sensitivity of the estimator to the removal of any single agent, which is particularly relevant to practical FL scenarios involving potential network outages, agent dropout due to constraints such as budget limitations or expired agreements, and poor local data quality. It also extends to settings where the learning algorithm operates under limited communication bandwidth.

**Lemma 4.11.** *The proposed Algorithm 1 satisfies agent-dependent stability with $\mu = O(\frac{1}{\sqrt{N}})$, where $N = Mn$ is the total number of samples.*

This stability result allows us to derive an out-of-sample generalization guarantee for our one-shot weighted ensemble estimator.

**Theorem 4.12** (Out-of-sample generalization bound). *Under Assumptions 4.1, 4.2, , 4.3 and at least $m' \geq 2$ agents have the same features, with quantile loss function $\ell(\cdot) = \rho_\tau(\cdot)$, we have*

$$\mathbb{E}\left[\ell\left(\widehat{\beta}^{OSW}(\tau), z\right)\right] - \frac{1}{N}\sum_{k=1}^{N}\ell\left(\widehat{\beta}^{OSW}(\tau), z_k\right) = O_p\left(\frac{1}{\sqrt{n}}\right).$$

These bounds show that the OSW estimator generalizes well to unobserved data and achieves the optimal estimation error convergence rate and the optimal generalization error convergence rate, which is consistent with the results in the single joint learning literature (Salehkaleybar et al., 2021).

## 5 NUMERICAL EXPERIMENTS

In this section, we evaluate the performance of the proposed estimator through simulations under various settings designed to illustrate the practical performance of our methods. We consider a data-generating process of the form: $Y_m = X_m\beta^\star(\tau) + \Xi_m, m = 1, \ldots, M$, where $X_m \in \mathbb{R}^{n \times d}$ is a matrix of covariates drawn from a multivariate normal distribution $\mathcal{N}(0, \Sigma)$, and $\Xi_m := (\xi_{1,m}, \cdots, \xi_{n,m})$ represents the noise vector. We evaluate the performance across three quantile levels $\{0.2, 0.5, 0.8\}$ with 4 different settings of the noise term: $\xi_{i,m}$ is generated from (a) standard normal $\mathcal{N}(0, 1)$, (b) heteroscedastic normal $\mathcal{N}(0, (2 + 0.1X_{i1})^2)$, (c) exponential $Exp(1)$, (d) $t$-distribution $t(5)$, and (e) Cauchy distribution.

We compare the one-shot weighted ensemble estimator (OSW) with the following baselines: (a) Naive-Local, which uses local estimates independently, (b) Naive One-shot Federated Learning (Naive-OSFL), which averages these local estimates, (c) Centralized, which uses a single machine to concentrate all data without missing features, providing an optimal baseline across algorithms, (d) UW-OSW, which replaces each $W_m$ with diagonal elements that are 1 and the remaining elements that follow a standard normal distribution to verify the optimality of $W_m$ within the algorithm framework, (f) DA-OSW, which examines the role of $T_m$ by substituting it with $I_d$. Table 2 summarizes the computation cost under these different methods. The performance is evaluated in terms of mean squared prediction error (MSPE) to assess out-of-sample performance. We further validate the asymptotic normality of our estimator by examining the convergence of its empirical distribution moments.

Table 2: Computation cost for agent $m$ under different methods

| Methods | Communication cost | Methods | Communication cost |
|---------|:------------------:|---------|:------------------:|
| Naive-Local | $--$ | Centralized | $--$ |
| Naive-OSFL | $O(d_i)$ | DA-OSW | $O(d_i^2)$ |
| UW-OSW | $O(d_i^2)$ | OSW | $O(d_i^2)$ |

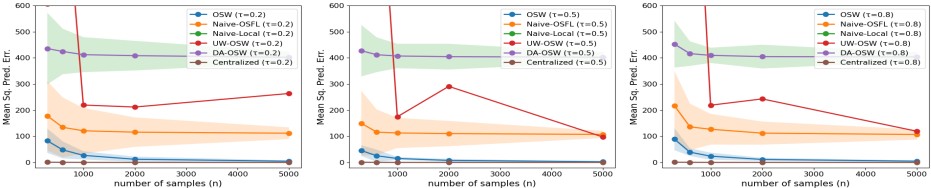

Figure 1: Mean squared prediction error under Cauchy distribution.

Due to space limitations, we present results only for Cauchy distributed noise setting. The results are shown in Figure 1, which empirically demonstrate superior performance of the proposed method,

along with the uniform convergence compared to baselines across different quantile levels. As the sample size increases, all approaches show the expected reduction in prediction error; however, the OSW method remains competitive with the centralized method and consistently achieves the lowest prediction error in all settings. In contrast, the other baseline methods exhibit limited improvement once the sample size exceeds 1000. In addition, as shown in Figure 2, the empirical mean and variance of the estimates converge to their theoretical values as the sample size increases, supporting the asymptotic normality of the estimator.

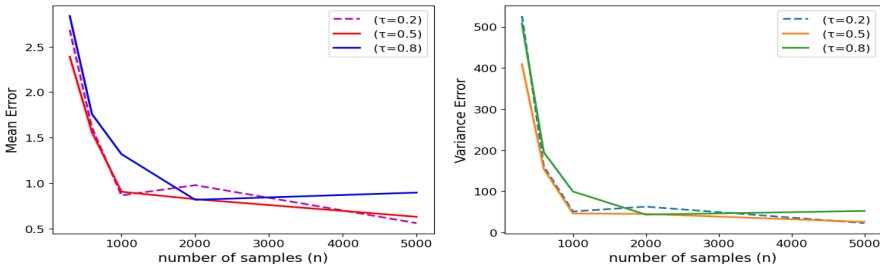

Figure 2: Convergence of the empirical mean and variance under the Cauchy distribution.

We emphasize that *consistent* findings are observed for various noise settings as previously mentioned, further demonstrating uniform performance guarantees and, particularly, the *robustness against* outliers and heavy-tailed noise. We refer readers to Appendix A for details on the setup and completed results of simulation experiments (Appendix A.1 - A.4) and real data analysis (Appendix A.6), particularly the sensitivity analysis under non-Gaussian settings and related discussion (Appendix A.5).

## 6 CONCLUSION

This paper presents a unified framework for federated quantile regression, tackling challenges from heterogeneous features and nonsmooth loss functions. The proposed one-shot weighted ensemble estimator avoids iterative communication while maintaining statistical efficiency. It is asymptotically normal, stable, and offers strong generalization guarantees under mild conditions. QR handles heavy-tailed or skewed distributions well, and our method retains this robustness in federated settings. Still, feature heterogeneity may affect aggregation efficiency. Establishing finite-sample guarantees under heavy-tailed conditions remains an important avenue for future research. Additionally, the current theory is limited by the Gaussian assumption. We emphasize, however, that establishing the theoretical guarantee, e.g., asymptotic normality of the learned parameters, and the determination of optimal weight matrix $W^\star$, remains technically challenging even under Gaussian features: unlike least squares, the local quantile regression estimator does not admit a closed-form expression, and the nonsmoothness of the quantile loss further complicates the analysis. Our results, therefore, require new techniques, such as Bahadur linear representation, beyond those used for federated mean regression. To the best of our knowledge, our work is the first to investigate federated quantile regression with such heterogeneous structured features. Therefore, as a starting point, we impose Gaussian design assumptions to keep the setting analytically tractable. We leave the work with more relaxed assumptions on features for future work.

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

The Appendix is organized as follows. Section A details the experimental setup and reports the complete results of the simulations and real data analysis. Section B contains the proofs of the main theoretical results presented in the paper.

## A  FULL EXPERIMENTS

All experiments were conducted on a Windows machine equipped with an i7-12700H (2.30 GHz) CPU, an NVIDIA 3070Ti GPU, and 32 GB of RAM. We consider a federated setting with $M = 40$ agents, each observing a subset of $d = 40$ features. Across different experiments, we set the sample size per agent to $n = 100, 1000, 2000, 5000$. The data is generated via the linear regression model $Y_m = X_m \beta^\star(\tau) + \Xi_m$, for $m = 1, \ldots, M$, where $X_m \in \mathbb{R}^{n \times d}$ is generated from $N(0, \Sigma)$, and $\Xi_m := (\xi_{1,m}, \ldots, \xi_{n,m})^\top$. Among the 40 agents, 10 observe random subsets of 30 features, while the remaining 30 observe random subsets of 25 features. To construct the covariance matrix $\Sigma$, we sample $d$ eigenvalues from the uniform distribution on $[0, 1]$, randomly amplify three of them by a factor of 30, and set $\Sigma = W \Lambda W^\top$, where $\Lambda$ is the diagonal matrix of eigenvalues and $W$ is a random orthogonal matrix. Figure 3 displays the heatmap of this covariance matrix. The noise term $\xi_{i,m}$ is generated under four scenarios: $\xi_{i,m}$ is generated from (a) homoscedastic normal $\mathcal{N}(0, 1)$, (b) heteroscedastic normal $\mathcal{N}(0, (2 + 0.1 X_{i1})^2)$, (c) exponential $Exp(1)$, and (d) $t$-distribution $t(5)$.

Denote $\beta(\tau) \in \mathbb{R}^d$ be a vector generated by drawing $d$ samples from $N(0, 10)$. For each quantile level $\tau$, we shift $\beta(\tau) \in \mathbb{R}^d$ such that noise term $\xi$ satisfying $\mathbb{P}(\xi_{i,m} \le 0 \mid X_i) = \tau$ to generate true coefficient $\beta^\star(\tau)$. Specifically, we consider the following scenarios, (a) Homoscedastic normal: $\beta^\star(\tau) = \beta(\tau) + \Phi^{-1}(\tau) e_1$, (b) Heteroscedastic normal: $\beta^\star(\tau) = \beta(\tau) + 2\Phi^{-1}(\tau) e_1 + 0.1\Phi^{-1}(\tau) e_2$, (c) Exponential: $\beta^\star(\tau) = \beta(\tau) + F_{\exp}^{-1}(\tau) e_1$, and (d) $t$: $\beta^\star(\tau) = \beta(\tau) + 5F_t^{-1}(\tau) e_1$, where $\Phi$ is the cumulative distribution function (CDF) of the standard normal distribution, $F_{\exp}$ and $F_t$ denote the CDFs of the exponential and $t$ distributions, respectively, and $e_t$ is the standard basis vector in $\mathbb{R}^d$ with the $t$th element being one and all the other elements being zero.

Throughout the numerical experiments, the key quantities $T_m$ and $W^\star$ are estimated by aggregating information from all agents. The density $f_{\xi_{i,m}}(0 \mid x_{i,m+})$ is estimated using a one-dimensional kernel density estimator based on the residuals $r_{i,m} = y_{i,m} - x_{i,m+}^\top \tilde{\beta}_m(\tau)$. Specifically,

$$\widehat{f}_{\xi_{i,m}}(0 \mid x_{i,m+}) = \frac{1}{n h_m} \sum_{i=1}^{n} K(r_{i,m}/h_m),$$

where $K(\cdot)$ is the Gaussian kernel, $K(u) = (2\pi)^{-1/2} \exp(-u^2/2)$. For the bandwidth $h_m$, we adopt Silverman's rule of thumb $h_m = 1.06 \hat{\sigma}_{r,m} n_m^{-1/5}$, where $\hat{\sigma}_{r,m}$ is the sample standard deviation of $\{r_{i.m}\}_{i=1}^n$, and $n$ is the sample size for agent $m$.

All methods are evaluated on a held-out test agent with access to all 40 features. Experiments are repeated across quantile levels $\tau = 0.2, 0.5$, and $0.8$.

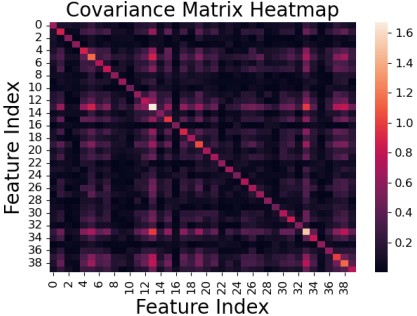

Figure 3: The heatmap of the covariance matrix $\Sigma$.

## A.1 HOMOSCEDASTIC NORMAL DISTRIBUTION

In this section, we present results for the classical symmetric case—the normal distribution, where $\xi_{i,m} \sim N(0,1)$. The results are shown in Figure 4 and Figures A.1. Figure 4 demonstrate superior performance of the proposed method under this setting, along with the uniform convergence compared to baselines across different quantile levels. As the sample size increases, all approaches show the expected reduction in prediction error; however, the OSW method remains competitive with the centralized method and consistently achieves the lowest prediction error across different quantile levels. In contrast, the other baseline methods exhibit limited improvement once the sample size exceeds 1000.

To validate the asymptotic normality of the proposed estimator $\widehat{\beta}^{\mathrm{OSW}}(\tau)$, we present the mean and variance of estimation errors, defined as $\widehat{\beta}^{\mathrm{OSW}}(\tau) - \beta^{\star}(\tau)$, in Figure A.1. The horizontal axis represents the sample size $n$, while the vertical axis shows the mean (left) and variance (right) of the estimation error. As $n$ increases, both the mean and variance decrease, empirically confirming our asymptotic normality results.

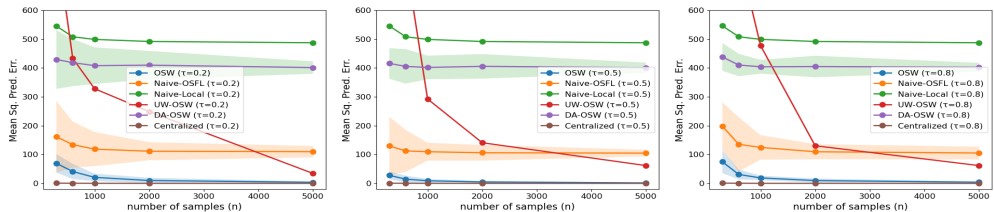

Figure 4: Mean squared prediction error under a homoscedastic normal distribution.

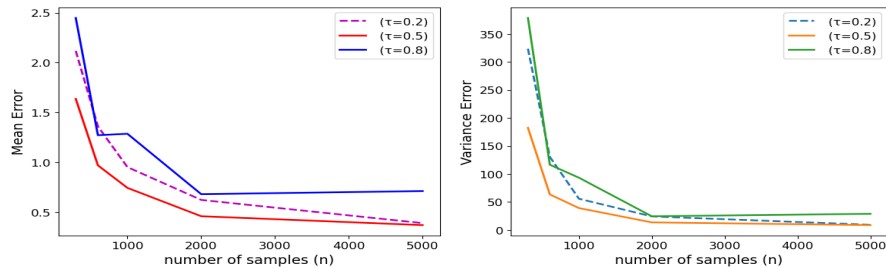

Figure 5: Convergence of the empirical mean and variance under a homoscedastic normal distribution.

## A.2 HETEROSCEDASTIC NORMAL DISTRIBUTION

Figures 6 and 7 illustrate the performance of our proposed estimator under a heteroscedastic normal distribution, where $\xi_{i,m} \sim N(0, (2 + 0.1X_{i1})^2)$. The results further demonstrate the superior performance of the proposed method in terms of prediction accuracy compared to other baseline methods.

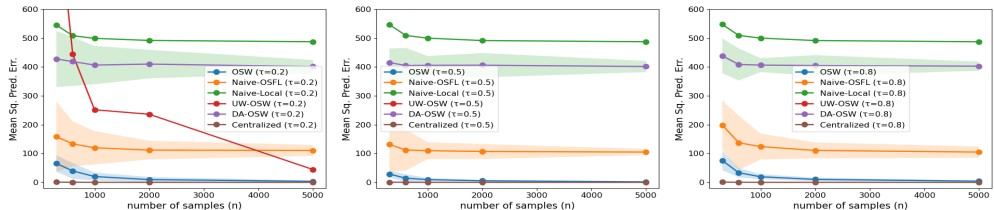

Figure 6: Mean squared prediction error under heteroscedastic normal distribution.

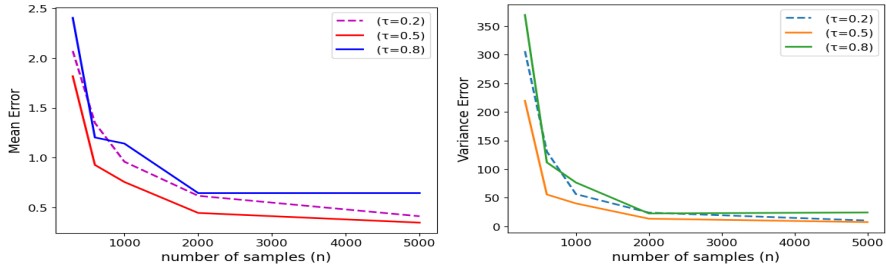

Figure 7: Convergence of the empirical mean and variance under heteroscedastic normal distribution.

## A.3 EXPONENTIAL DISTRIBUTION

Figure 8 and Figure 9 present the results under exponential distribution across different quantile levels.

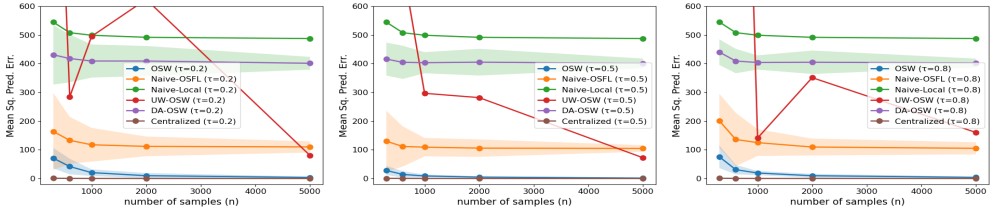

Figure 8: Mean squared prediction error under exponential distribution exp(1).

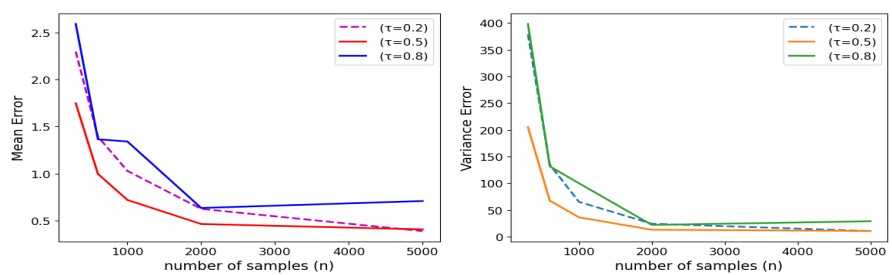

Figure 9: Convergence of the empirical mean and variance under exponential distribution.

## A.4 STUDENT-T DISTRIBUTION

Figure 10 and Figure 11 present the results under the $t(5)$ distribution across different quantile levels.

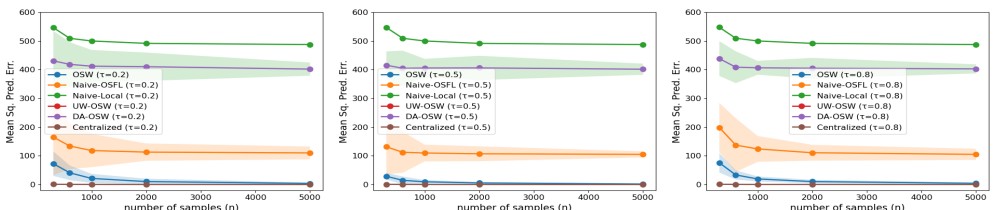

Figure 10: Mean squared prediction error under $t(5)$ distribution.

To summarize, all of these results consistently demonstrate superior prediction performance of our proposed method across various noise settings and quantile levels. The proposed method exhibits

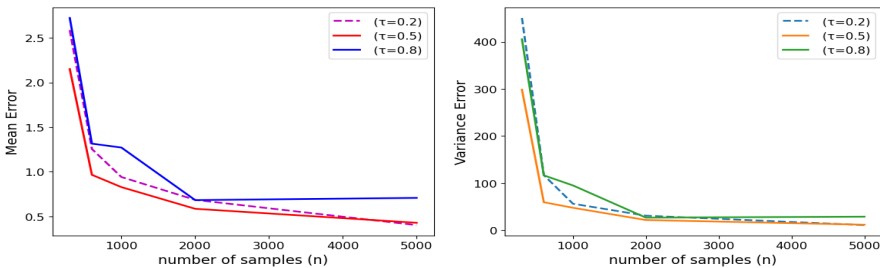

Figure 11: Convergence of the empirical mean and variance under $t(5)$ distribution.

robustness to outliers and heavy-tailed noise distributions. Notably, our proposed method remains competitive with the centralized method.

### A.5 GENERALIZATION BEYOND GAUSSIAN ASSUMPTIONS: SENSITIVITY ANALYSIS AND DISCUSSION

As discussed in Remark 4.6, the current theory relies on the Gaussianity assumption. Relaxing this assumption would be both valuable and novel, but it also poses significant technical challenges, particularly in establishing strong theoretical guarantees and deriving optimal weight estimators: Many theoretical properties and independence structures that hold under Gaussianity may no longer be valid in non-Gaussian settings. For instance, extending the framework to sub-Gaussian designs introduces new technical challenges, as several simplifications enabled by Gaussianity break down.

In this section, we examine the sensitivity of our approach to this assumption by conducting experiments on non-Gaussian data. In the subsequent simulation study, we assume the data are distributed according to the $t$ and exponential distributions, and we set the quantile level at $\tau = 0.5$ for the FQR model. The results are reported in Tables 3 and 4, corresponding to the $t$ and exponential distributions, respectively.

Table 3: MSPE under different sample sizes when data is generated from the $t$ distribution.

| $N$ | Naive-OSFL | Naive-Local | OSW |
|---|---|---|---|
| 500 | 86.91 | 534.96 | 17.21 |
| 1,000 | 91.95 | 510.86 | 9.18 |
| 2,000 | 86.17 | 495.26 | 2.92 |
| 5,000 | 87.48 | 499.84 | 2.06 |

Table 4: MSPE under different sample sizes when data is generated from the exponential distribution.

| $N$ | Naive-OSFL | Naive-Local | OSW |
|---|---|---|---|
| 500 | 149.05 | 508.40 | 45.89 |
| 1,000 | 125.39 | 518.92 | 30.19 |
| 2,000 | 127.73 | 512.78 | 24.03 |
| 5,000 | 123.99 | 510.27 | 21.40 |

The tables show that our proposed algorithm (OSW) consistently attains the lowest MSPE, with performance improving as sample size increases, demonstrating both adaptability to diverse data-generating processes and robustness in estimation.

We emphasize that these preliminary results suggest our method may generalize beyond the Gaussian setting. To relax the Gaussianity assumption, a feasible direction is to leverage tools such as linear projection techniques, matrix concentration inequalities, and uniform laws of large numbers to develop appropriate corrections and establish rigorous theoretical guarantees. We acknowledge the novelty and importance of this extension and leave it for future research.

## A.6 REAL DATA ANALYSIS: CALIFORNIA HOUSING PRICES

In this section, we illustrate the practical implementation of the proposed estimator using the California Housing dataset (https://lib.stat.cmu.edu/), which contains 1990 U.S. Census data on housing districts, including median income, average number of rooms, occupancy, and geographic coordinates. This dataset is widely used as a benchmark in statistics and machine learning for testing new methodologies. Our study includes 20 agents, where the first 10 lack the first dimension, and the remaining 10 lack seven dimensions. We examine how the mean squared prediction error (MSPE) varies with the local sample size $n$. For the FQR model, we set the quantile level to $\tau = 0.2, 0.5, 0.8$, and the results are summarized below, showing that our algorithm consistently achieves the best performance than other methods and remains competitive with the centralized method. MSPE decreases as $n$ increases, providing empirical support for the theoretical guarantees established in our work.

Table 5: Mean Squared Prediction Error (MSPE) for different sample sizes in real data($\tau = 0.2$)

| $n$ | Naive-OSFL | Naive-Local | OSW | DA-OSW | UW-OSW | Centralized |
|-----|-----------|-------------|-----|--------|--------|-------------|
| 10  | 1891.69   | 6257.71     | **8.10** | 4346.21 | 368.42  | 5.41 |
| 20  | 12.51     | 3750.04     | 8.10 | 1560.21 | 3210.45 | 5.20 |
| 30  | 432.72    | 5540.64     | **8.10** | 3218.72 | 1239.43 | 4.61 |
| 350 | 439.30    | 1687.57     | **8.10** | 573.21  | 84.26   | 3.30 |
| 450 | 21.05     | 487.95      | **4.49** | 287.62  | 66.45   | 2.18 |
| 500 | 21.06     | 557.38      | **3.89** | 163.43  | 50.67   | 2.08 |

Table 6: Mean Squared Prediction Error (MSPE) for different sample sizes in real data($\tau = 0.5$)

| $n$ | Naive-OSFL | Naive-Local | OSW | DA-OSW | UW-OSW | Centralized |
|-----|-----------|-------------|-----|--------|--------|-------------|
| 10  | 4442.90   | 6257.72     | **8.13** | 7321.50 | 5324.31 | 5.46 |
| 20  | 16.86     | 2312.25     | 8.10 | 5321.89 | 3210.45 | 5.20 |
| 30  | 432.72    | 5540.64     | **8.10** | 3291.26 | 1565.36 | 4.68 |
| 350 | 42.76     | 844.46      | **6.50** | 623.45  | 98.72   | 3.28 |
| 450 | 23.01     | 518.64      | **4.59** | 320.08  | 69.17   | 2.14 |
| 500 | 44.42     | 575.89      | **4.40** | 198.76  | 54.69   | 2.03 |

Table 7: Mean Squared Prediction Error (MSPE) for different sample sizes in real data($\tau = 0.8$)

| $n$ | Naive-OSFL | Naive-Local | OSW | DA-OSW | UW-OSW | Centralized |
|-----|-----------|-------------|-----|--------|--------|-------------|
| 10  | 4106.56   | 2243.31     | **8.06** | 3278.56 | 4326.78 | 5.34 |
| 20  | 54.72     | 13260.42    | 8.05 | 2654.21 | 3769.08 | 5.06 |
| 30  | 234.03    | 7299.73     | 8.05 | 1856.23 | 827.36  | 4.51 |
| 350 | 80.73     | 1076.19     | **6.68** | 467.24  | 56.45   | 3.45 |
| 450 | 67.86     | 912.26      | **6.17** | 187.48  | 31.85   | 2.11 |
| 500 | 44.85     | 579.21      | **5.29** | 164.36  | 23.19   | 1.99 |

## B  THEORETICAL PROOFS

### B.1  PROOFS FOR LEMMA 4.4

*Proof.* Let $\beta^\star(\tau) = [\beta_+^\star(\tau)^\top, \beta_-^\star(\tau)^\top]^\top \in \mathbb{R}^d$ be the global parameter vector. The response variable is generated as:

$$Y_m = X_{m+}^\top \beta_+^\star(\tau) + X_{m-}^\top \beta_-^\star(\tau) + \Xi_m,$$

where the error term $\Xi_m$ satisfies the quantile condition $Q_\tau(\Xi_m | X_{m+}, X_{m-}) = 0$. Recall that the quantile loss function is given by $\rho_\tau(u) = u(\tau - I(u < 0))$, and the local QR estimator for agent $m$ is:

$$\tilde{\beta}_m(\tau) = \underset{\beta(\tau) \in \mathbb{R}^d}{\arg\min} \widehat{Q}_m(\beta(\tau)), \quad \text{where } \widehat{Q}_m(\beta(\tau)) =: \frac{1}{n} \sum_{i=1}^n \rho_\tau \left( y_{i,m} - (x_{i,m+})^\top \beta(\tau) \right).$$

Note that the expected loss function has the form $\beta_m^\star(\tau) = \underset{\beta(\tau) \in \mathbb{R}^d}{\arg\min} Q_m(\beta(\tau))$, where $Q_m(\beta(\tau)) = \mathbb{E}\left[ \rho_\tau(y_{i,m} - x_{i,m+}^\top \beta(\tau)) \right]$. Substituting $y_{i,m}$ with the true model yields that

$$Q_m(\beta(\tau)) = \mathbb{E}\left[ \rho_\tau \left( x_{i,m+}^\top(\beta_+^\star(\tau) - \beta(\tau)) + x_{i,m-}^\top \beta_-^\star(\tau) + \xi_{i,m} \right) \right]. \tag{9}$$

The first-order condition for minimization (9) requires:

$$\frac{\partial Q_m(\beta(\tau))}{\partial \beta(\tau)} = \mathbb{E}\left[ \psi_\tau \left( x_{i,m+}^\top(\beta_+^\star(\tau) - \beta(\tau)) + x_{i,m-}^\top \beta_-^\star(\tau) + \xi_{i,m} \right) x_{i,m+} \right] = 0, \tag{10}$$

where $\psi_\tau(u) = \tau - \mathbb{I}(u < 0)$. Because $x_i$ are Gaussian random vectors, it follows that $x_{i,m-} = \Sigma_{i\pm}^\top \Sigma_{i+}^{-1} x_{i,m+} + \mathbf{v}$, $\mathbf{v} \sim N(0, \Gamma_{i-})$, where $\Gamma_{i-} = \Sigma_{i-} - \Sigma_{i\pm}^\top \Sigma_{i+}^{-1} \Sigma_{i\pm}$ is the Schur complement. After Taylor expansion of the probability term and simplification, we obtain

$$\mathbb{E}\left[ f_{\xi_m}(0 | x_{i,m+}) x_{i,m+} x_{i,m+}^\top \left( \beta_+^\star(\tau) - \beta(\tau) + \Sigma_{i+}^{-1} \Sigma_{i\pm} \beta_-^\star(\tau) \right) \right] = 0. \tag{11}$$

The local optimal parameter is then given by

$$\beta_m^\star(\tau) = \beta_+^\star(\tau) + \underbrace{\left( \mathbb{E}\left[ f_{\xi_{i,m}}(0 | x_{i,m+}) x_{i,m+} x_{i,m+}^\top \right] \right)^{-1} \mathbb{E}\left[ f_{\xi_{i,m}}(0 | x_{i,m+}) x_{i,m+} x_{i,m-}^\top \right]}_{A^{-1}B} \beta_-^\star(\tau).$$

$$\tag{12}$$

Define the projection matrix:

$$T_m := \begin{bmatrix} I_{d_m} & A^{-1}B \end{bmatrix} \Pi_m,$$

where:

- $A = \mathbb{E}\left[ f_{\xi_{i,m}}(0 | x_{i,m+}) x_{i,m+} x_{i,m+}^\top \right]$ is the weighted Hessian matrix,

- $B = \mathbb{E}\left[ f_{\xi_{i,m}}(0 | x_{i,m+}) x_{i,m+} x_{i,m-}^\top \right]$ is the weighted covariance matrix,

- $\Pi_m$ is a feature permutation matrix.

Under the Assumption 4.1, by the Glivenko-Cantelli theorem, the sample loss function converges uniformly to the expected loss function, we have

$$\sup_{\beta(\tau)} \left| \frac{1}{n} \sum_{i=1}^n \rho_\tau \left( y_{i,m} - x_{i,m+}^\top \beta(\tau) \right) - Q_m(\beta(\tau)) \right| \xrightarrow{p} 0. \tag{13}$$

As the expected loss function $Q_m(\beta(\tau))$ is strictly convex at $\beta_m^\star(\tau)$ (since the Hessian matrix $A \succ 0$), the existence of a unique minimum exists. According to the limit theorem, we have $\tilde{\beta}_m(\tau) \xrightarrow{p} T_m \beta^\star(\tau)$.

Convergence in probability follows from the uniform law of large numbers under standard regularity conditions. Substituting back into $\widehat{\beta}(\tau; \Omega(W))$, we can apply the continuous mapping theorem to

derive that

$$
\begin{aligned}
\widehat{\beta}(\tau; \Omega(W)) &= \left( \sum_{m=1}^{M} T_m^\top W_m T_m \right)^{-1} \left( \sum_{m=1}^{M} T_m^\top W_m \tilde{\beta}_m(\tau) \right) \\
&\xrightarrow{p} \left( \sum_{m=1}^{M} T_m^\top W_m T_m \right)^{-1} \left( \sum_{m=1}^{M} T_m^\top W_m T_m \beta^\star(\tau) \right) = \beta^\star(\tau).
\end{aligned}
$$

$\square$

### B.2 Proof for Theorem 4.5

*Proof.* Note that the expected loss function is given by

$$
\beta_m^\star(\tau) = \underset{\beta(\tau) \in \mathbb{R}^d}{\arg\min} Q_m(\beta(\tau)), \qquad \text{where } Q_m(\beta(\tau)) =: \mathbb{E}\left[ \rho_\tau(y_{i,m} - x_{i,m+}^\top \beta(\tau)) \right],
$$

where the local parameter vector $\beta_m^\star(\tau) \in \mathbb{R}^{d_m}$ connects to a global parameter vector $\beta^\star(\tau) \in \mathbb{R}^d$ through a $m$-dependent projection matrix $T_m \in \mathbb{R}^{d_m \times d}$, which can be formalized as

$$
\beta_m^\star(\tau) = T_m \beta^\star(\tau), \quad \forall m \in \{1, \dots, M\}.
$$

The local model's error term, capturing the deviation between observed responses and their conditional quantile predictions, is formally defined as:

$$
\xi_{i,m}(\tau) := y_{i,m} - x_{i,m+}^\top \beta_m^\star(\tau),
$$

and the local QR estimator for agent $m$ is obtained as

$$
\tilde{\beta}_m(\tau) = \underset{\beta \in \mathbb{R}^d}{\arg\min} \widehat{Q}_m(\beta(\tau)), \quad \text{where } \widehat{Q}_m(\beta(\tau)) =: \frac{1}{n} \sum_{i=1}^{n} \rho_\tau \left( y_{i,m} - (x_{i,m+})^\top \beta(\tau) \right).
$$

For each observation $m$, define the weighted residual $r_{i,m}(\beta(\tau)) = \tau - I[y_{i,m} - (x_{i,m+})^\top \beta(\tau) < 0]$. The first derivative of the loss function is:

$$
S_{n,m}(\beta_m(\tau)) = \frac{\partial \widehat{Q}_m(\beta(\tau))}{\partial \beta(\tau)} = -\frac{1}{n} \sum_{i=1}^{n} x_{i,m+} r_{i,m}(\beta(\tau)).
$$

Setting the derivative equal to zero, we obtain the equation that the estimator $\tilde{\beta}_m(\tau)$ must satisfy:

$$
\frac{1}{n} \sum_{i=1}^{n} x_{i,m+} r_{i,m}(\tilde{\beta}_m(\tau)) = 0.
$$

Expanding the first-order derivative around the true parameter $\beta_m^\star(\tau)$ using a Taylor series, we get:

$$
\frac{1}{n} \sum_{i=1}^{n} x_{i,m+} r_{i,m}(\tilde{\beta}_m(\tau)) \approx \frac{1}{n} \sum_{i=1}^{n} x_{i,m+} r_{i,m}(\beta_m^\star(\tau)) + \frac{1}{n} \sum_{i=1}^{n} x_{i,m+} (\tilde{\beta}_m(\tau) - \beta_m^\star(\tau)) \frac{\partial r_{i,m}(\beta_m^\star(\tau))}{\partial \beta(\tau)}.
$$

Since $\tilde{\beta}_m(\tau)$ satisfies $\frac{1}{n} \sum_{i=1}^{n} x_{i,m+} r_{i,m}(\tilde{\beta}_m(\tau)) = 0$, this simplifies to

$$
\sum_{i=1}^{n} x_{i,m+} (\tilde{\beta}_m(\tau) - \beta_m^\star(\tau)) \frac{\partial r_{i,m}(\beta_m^\star(\tau))}{\partial \beta} \approx -\sum_{i=1}^{n} x_{i,m+} r_{i,m}(\beta_m^\star(\tau)).
$$

We next compute the expectations of $\frac{\partial r_{i,m}(\beta_m^\star(\tau))}{\partial \beta(\tau)}$ and $r_{i,m}(\beta_m^\star(\tau))$.

As $\beta_m^\star(\tau)$ is the true parameter for the $\tau$-th quantile, $E[r_{i,m}(\beta_m^\star(\tau))] = 0$, $\frac{\partial r_{i,m}(\beta_m^\star(\tau))}{\partial \beta(\tau)} = -f_{\xi_{i,m}}(0|x_{i,m+}) x_{i,m+}$, where $f_{\xi_{i,m}}(0|x_{i,m+})$ is the density of the error term $\xi_{i,m}$ at 0 given $x_{i,m+}$. Under assumption 4.2, working conditional on the design $\mathcal{F}_X = \sigma\{x_{i,m+}\}_{i=1}^{n}$ and using Knight's identity, we obtain the stochastic linearization

$$
0 = S_{n,m}(\beta_m^\star(\tau)) - A_{n,m}(\tilde{\beta}_m(\tau) - \beta_m^\star(\tau)) + r_{n,m}, \tag{14}
$$

where

$$A_{n,m} := \frac{1}{n} \sum_{i=1}^{n} E_{\xi_{i,m}} \left[ \frac{\partial r_{i,m}(\beta_m^\star(\tau))}{\beta(\tau)} \Big| \mathcal{F}_X \right] = \frac{1}{n} \sum_{i=1}^{n} f_{\xi_{i,m}}(0 \mid x_{i,m+}) \, x_{i,m+} x_{i,m+}^\top, \qquad (15)$$

$$r_{n,m} = o_p\big(\|\tilde{\beta}_m(\tau) - \beta_m^\star(\tau)\|\big) + o_p(n^{-1/2}) \qquad (\text{given } \mathcal{F}_X).$$

Thus, the expectation is taken only over the label noise in the Jacobian evaluated at the *fixed* point $\beta_m^\star$; we *do not* take an expectation of the entire first-order condition. A conditional LLN yields

$$A_{n,m} = A_m + o_p(1), \qquad A_m = \mathbb{E}\left[ f_{\xi_{i,m}}(0|x_{i,m+}), x_{i,m+} x_{i,m+}^\top \right]. \qquad (16)$$

Combining equation 14–equation 16 gives the Bahadur representation

$$A_m \left( \tilde{\beta}_m - \beta_m^\star \right) = \frac{1}{n} \sum_{i=1}^{n} x_{i,m+} r_{i,m}(\beta^\star(\tau)) + o_p(n^{-1/2}). \qquad (17)$$

Equivalently, keeping the *random* Jacobian leads to

$$\sqrt{n} \left( \tilde{\beta}_m(\tau) - \beta_m^\star(\tau) \right) = A_{n,m}^{-1} \left( \frac{1}{\sqrt{n}} \sum_{i=1}^{n} x_{i,m+} r_{i,m}(\beta^\star(\tau)) \right) + o_p(1).$$

Since $A_{n,m}^{-1} - A_m^{-1} = o_p(1)$, Slutsky's theorem implies the same $\sqrt{n}$-limit if $A_{n,m}$ is replaced by $A_m$. Hence, the randomness induced by the noise is fully preserved; the impact of $\xi_{i,m}$ on the Jacobian is $o_p(1)$ and absorbed in the remainder. This implies that

$$\sqrt{n}(\tilde{\beta}_m(\tau) - \beta_m^\star(\tau)) \approx \left[ E[f_{\xi_{i,m}}(0|x_{i,m+}) x_{i,m+} x_{i,m+}^\top] \right]^{-1} \cdot \frac{1}{\sqrt{n}} \sum_{i=1}^{n} x_{i,m+} r_{i,m}(\beta_m^\star(\tau)).$$

By the Central Limit Theorem (CLT), as the sample size $n$ approaches infinity, the distribution of the sample mean converges to a normal distribution. For our case:

$$\sqrt{n} \left( \frac{1}{n} \sum_{i=1}^{n} x_{i,m+} r_{i,m}(\tilde{\beta}_m(\tau)) \right) \xrightarrow{d} N(0, V),$$

where $V = \text{Var}(x_{i,m+} r_{i,m} \beta_m^\star(\tau))$ is the variance of $x_{i,m+} r_{i,m}(\beta_m^\star(\tau))$. Therefore,

$$\sqrt{n}(\tilde{\beta}_m(\tau) - \beta_m^\star(\tau)) \xrightarrow{d} N(0, \psi_m). \qquad (18)$$

The covariance matrix $\psi_m$ can be calculated using the following formula:

$$\psi_m = \left[ E[f_{\xi_{i,m}}(0|x_{i,m+}) x_{i,m+} x_{i,m+}^\top] \right]^{-1} \cdot V \cdot \left[ E[f_{\xi_{i,m}}(0|x_{i,m+}) x_{i,m+} x_{i,m+}^\top] \right]^{-1}.$$

Specifically, $V = \text{Var}(x_{i,m+} r_{i,m} \beta_m^\star(\tau))$ can be written as:

$$V = E[(x_{i,m+} r_{i,m}(\beta_m^\star(\tau)))(x_{i,m+} r_{i,m}(\beta_m^\star(\tau)))^\top].$$

Noting that $r_{i,m}(\beta_m^\star(\tau)) = \tau - I[y_{i,m} - x_{i,m+}^\top \beta_m^\star(\tau) < 0]$, we have $r_{i,m}(\beta_m^\star(\tau))^2 = \tau(1 - \tau)$. Finally, the covariance matrix $\psi_m$ is expressed as

$$\psi_m = \tau(1 - \tau) \left[ E[f_{\xi_{i,m}}(0|x_{i,m+}) x_{i,m+} x_{i,m+}^\top] \right]^{-1} \cdot \Sigma_{i+} \cdot \left[ E[f_{\xi_{i,m}}(0|x_{i,m+}) x_{i,m+} x_{i,m+}^\top] \right]^{-1}.$$

We proceed to show $C(W_1, \cdots, W_M) \succeq C^*$ under general feature distribution $P$ and $W_m^\star := \psi_m^{-1}$. Based on the form $\widehat{\beta}(\tau; \Omega(W)) = \left( \sum_{m=1}^{M} T_m^\top W_m T_m \right)^{-1} \left( \sum_{m=1}^{M} T_m^\top W_m \tilde{\beta}_m(\tau) \right)$, it follows that

$$\sqrt{n} \left( \widehat{\beta}(\tau; \Omega(W)) - \beta^\star(\tau) \right) \sim N(0, C(W_1, \cdots, W_M)), \qquad (19)$$

where

$$C(W_1, \cdots, W_M) = \left( \sum_{m=1}^{M} T_m^\top W_m T_m \right)^{-1} \cdot \left( \sum_{m=1}^{M} T_m^\top W_m W_m^{*-1} W_m T_m \right) \cdot \left( \sum_{m=1}^{M} T_m^\top W_m T_m \right)^{-1}.$$

Under the oracle weighting configuration $W_m = W_m^\star$, we attain the theoretical minimum asymptotic covariance bound. The resultant covariance structure simplifies to:

$$C(W_1, \cdots, W_M) = \left( \sum_{m=1}^M T_m^\top W_m^\star T_m \right)^{-1}, \tag{20}$$

establishing the pivotal semi-definite inequality requiring verification:

$$C(W_1, \cdots, W_M) \succeq \left( \sum_{m=1}^M T_m^\top W_m T_m \right)^{-1} = C^*. \tag{21}$$

To establish this ordering, we introduce the key matrix decomposition:

$$H_m = \begin{bmatrix} T_m^\top W_m^\star T_m & T_m^\top W_m T_m \\ T_m^\top W_m T_m & T_m^\top W_m W_m^{*-1} W_m T_m \end{bmatrix} = \begin{bmatrix} T_m^\top W_m^{-\frac{1}{2}} \\ T_m^\top W_m W_m^{-\frac{1}{2}} \end{bmatrix} \begin{bmatrix} T_m^\top W_m^{-\frac{1}{2}} \\ T_m^\top W_m W_m^{-\frac{1}{2}} \end{bmatrix}^\top \succeq 0.$$

where the outer product formulation explicitly guarantees positive semi-definiteness. Aggregating these components yields:

$$\sum_{m=1}^M H_m = \begin{bmatrix} C^{*-1} & \sum_{m=1}^M T_m^\top W_m T_m \\ \sum_{m=1}^M T_m^\top W_m T_m & \sum_{m=1}^M T_m^\top W_m (W_m^\star)^{-1} W_m T_m \end{bmatrix} \succeq 0.$$

The Schur complement analysis of this block matrix yields:

$$0 \preceq C^{*-1} - \left( \sum_{m=1}^M T_m^\top W_m T_m \right) \left( \sum_{m=1}^m T_m^\top W_m W_m^{*-1} W_m T_m \right)^{-1}$$

$$\times \left( \sum_{m=1}^M T_m^\top W_m T_m \right) = C^{*-1} - C(W_1, \ldots, W_m)^{-1}.$$

$\square$

### B.3 PROOF OF LEMMA 4.7

*Proof.* Slutsky's theorem can be applied to show $\widehat{W}_m \xrightarrow{P} W_m^\star$, where

$$\widehat{W}_m = \widehat{R}_m \left( \widehat{V}_m \widehat{V}_m^\top \right)^{-1} \widehat{R}_m.$$

Note that

$$\widehat{\Sigma}_{m+} = \frac{1}{n} \sum_{i=1}^n \frac{x_{i,m+}^\top x_{i,m+}}{n} \xrightarrow{p} \Sigma_{i+},$$

$$\widehat{R}_m = \frac{1}{n} \sum_{i=1}^n [f_{\xi_{i,m}}(0|x_{i,m+}) x_{i,m+} x_{i,m+}^\top] \xrightarrow{p} \left[ E[f_{\xi_{i,m}}(0|x_{i,m+}) x_{i,m+} x_{i,m+}^\top] \right].$$

In addition,

$$\widehat{V}_m = \frac{1}{n} \sum_{i=1}^n x_{i,m+} r_{i,m}(\tilde{\beta}_m(\tau)),$$

implies that

$$\widehat{V}_m \cdot (\widehat{V}_m)^\top \xrightarrow{p} E[(x_{i,m+} r_{i,m}(\beta_m^\star(\tau)))(x_{i,m+} r_{i,m}(\beta_m^\star(\tau)))^\top].$$

$\square$

### B.4 PROOF OF THEOREM 4.8

*Proof.* We first prove asymptotic normality of $\sqrt{n}(\widehat{\beta}^{\text{OSW}}(\tau) - \widehat{\beta}(\tau)) \xrightarrow{d} \mathsf{N}(0, C(\Omega(W^\star)))$. We point out that Theorem 4.5 is not directly applicable, as we use estimated weights that reuse the training data. Note that the estimator can be decomposed as:

$$\sqrt{n}\left(\widehat{\beta}^{\text{OSW}}(\tau) - \beta^\star(\tau)\right) = \left(\sum_{m=1}^{M} T_m^\top \widehat{W}_m T_m\right)^{-1} \left(\sum_{m=1}^{M} T_m^\top \widehat{W}_m (\tilde{\beta}_m(\tau) - T_m \beta^\star(\tau))\right). \quad (22)$$

With the asymptotic normality established for $\sqrt{n}(\tilde{\beta}_m(\tau) - T_m \beta^\star(\tau))$ in Eq. equation 18, Slutsky's theorem and continuous mapping theorem, we can conclude that $\sqrt{n}(\widehat{\beta}^{\text{OSW}}(\tau) - \beta^\star(\tau)) \xrightarrow{d} \mathsf{N}(0, C(\Omega(W^\star)))$.

Applying the delta method to the mapping $\tilde{\beta}_m(\tau) \mapsto T_m \beta^\star(\tau)$, which maps from $\mathbb{R}^d$ to $\mathbb{R}^{d_m}$, immediately yields the asymptotic normality of $\widehat{\beta}_m^{\text{OSW}}(\tau)$ based on $\widehat{\beta}(\tau; \Omega(W))$. It remains to verify the inequality $T_m C^* T_m^\top \preceq W_m^{*-1}$.

To this end, observe that the difference $W_m^{*-1} - T_m C^* T_m^\top$ corresponds to the Schur complement of the block matrix

$$H = \begin{bmatrix} W_m^{*-1} & T_m \\ T_m^\top & C^{*-1} \end{bmatrix}.$$

Hence, it suffices to show that $H \succeq 0$. Using the identity

$$C^* = \left(\sum_{m=1}^{M} T_m^\top W_m^* T_m\right)^{-1},$$

we rewrite $H$ as

$$H = \begin{bmatrix} W_m^{*-1} & T_m \\ T_m^\top & \sum_{m=1}^{M} T_m^\top W_m^* T_m \end{bmatrix} \succeq \begin{bmatrix} W_m^{*-1} & T_m \\ T_m^\top & T_m^\top W_m^* T_m \end{bmatrix}.$$

The right-hand side is clearly positive semidefinite since it can be expressed as a Gram matrix:

$$\begin{bmatrix} W_m^{*-\frac{1}{2}} \\ T_m^\top W_m^{*\frac{1}{2}} \end{bmatrix} \begin{bmatrix} W_m^{*-\frac{1}{2}} \\ T_m^\top W_m^{*\frac{1}{2}} \end{bmatrix}^\top \succeq 0.$$

This completes the proof.

$\square$

### B.5 PROOF OF COROLLARY 4.9

*Proof.* According to $\sqrt{n}(\widehat{\beta}^{\text{OSW}}(\tau) - \widehat{\beta}(\tau)) \xrightarrow{d} \mathsf{N}(0, C(\Omega(W^\star)))$, we can directly get

$$\left\|\widehat{\beta}^{\text{OSW}}(\tau) - \beta^\star(\tau)\right\| = O_p\left(\frac{1}{\sqrt{n}}\right).$$

$\square$

### B.6 PROOF OF THEOREM 4.12

*Proof.* When agent $m$ is removed from the distributed system, its $n$ samples $Z_m$ are uniformly redistributed to $m'$ compatible agents (Assumption (4.2)). Each compatible agent $j \in I_m$ receives $\Delta n = \frac{n}{m'}$ samples, updating its local sample size to

$$n_j = n \cdot \left(1 + \frac{1}{m'-1}\right).$$

Here, $Z_m$ denotes the set of data on the deleted agent and $I_m$ denotes the set of compatible agents.

The original local estimator on agent $j$, before removal, admits the Bahadur representation

$$\tilde{\beta}_j(\tau) = \beta_j^\star(\tau) + \frac{1}{n} D_j^{-1} \sum_{i=1}^n x_{i,j+} \left(\tau - I\{y_{i,j} < (x_{i,j+})^\top \beta_j^\star(\tau)\}\right) + o_p(n^{-1/2}),$$

where $D_j = \mathbb{E}\left[f_{Y|X}(x_{i,j+}^\top \beta_j^\star(\tau)) x_{i,j+} x_{i,j+}^\top\right] \succeq \underline{f} \cdot \mathbb{E}[x_{i,j+} x_{i,j+}^\top]$ (Assumption (4.3)). After redistributing $\Delta n$ samples, the updated estimator becomes

$$\tilde{\beta}_j^{\text{new}}(\tau) = \beta_j^\star(\tau) + \frac{1}{n_j} D_j^{-1} \left(\sum_{i=1}^n x_{i,j+} \left(\tau - I\{y_{i,j} < (x_{i,j+})^\top \beta_j^\star(\tau)\}\right)\right)$$

$$+ \frac{1}{n_j} D_j^{-1} \left(\sum_{k=1}^{n_j - n} x_{k,j+} \left(\tau - I\{y_{k,j} < (x_{k,j+})^\top \beta_j^\star(\tau)\}\right)\right) + o_p(n^{-1/2}).$$

The discrepancy between the original and updated estimators on compatible agents $j \in I_m$ is characterized by

$$\tilde{\beta}_j(\tau) - \tilde{\beta}_j^{\text{new}}(\tau) = \underbrace{\left(\frac{1}{n} - \frac{1}{n_j}\right) D_j^{-1} \sum_{i=1}^n x_{i,j+} (\tau - I\{\cdot\})}_{(A)} - \underbrace{\frac{1}{n_j} D_j^{-1} \sum_{k \in \tilde{Z}_j^{(k)}} x_{k,j+} (\tau - I\{\cdot\})}_{(B)} + o_p(n^{-1/2}),$$

(23)

where $\tilde{Z}_j^{(k)}$ contains $\Delta n = n_j - n = \frac{n}{m'}$ independent samples.

For $(A)$, we have

$$\left|\frac{1}{n} - \frac{1}{n_j}\right| = \left|\frac{1}{n} - \frac{1}{n\left(1 + \frac{1}{m'-1}\right)}\right| = \frac{1}{n} \cdot \frac{1}{m'+1} = O\left(\frac{1}{nm'}\right). \quad (24)$$

Here, we define

$$Z_A := \sum_{i=1}^n x_{i,j+} \cdot r_{i,j}(\beta_j^\star(\tau)), \quad \text{where} \quad r_{i,j}(\beta_j^\star(\tau)) = \tau - I\left[y_{i,j} - (x_{i,j+})^\top \beta_j^\star(\tau) < 0\right].$$

Since samples are independently and identically distributed, we know that the mean $E[r_{i,j}(\beta_j^\star(\tau))] = 0$ and the variance $\text{Var}(Z_A) = n \cdot \mathbb{E}\left[x_{i,j+} x_{i,j+}^\top \cdot \text{Var}(r_{i,j}(\beta_j^\star(\tau)) \mid x_{i,j+})\right] = n \cdot \tau(1-\tau) \cdot \mathbb{E}[x_{i,j+} x_{i,j+}^\top]$. Therefore, by the Central Limit Theorem:

$$\frac{1}{\sqrt{n}} Z_A \xrightarrow{d} \text{N}(0, \tau(1-\tau) \cdot \mathbb{E}[x_{j+} x_{j+}^\top]).$$

Therefore, the order of the original sum is:

$$Z_A = O_p(\sqrt{n}). \quad (25)$$

Bringing equations (24), (25) into (23), we have

$$(A) = \left(\frac{1}{n} - \frac{1}{n_j}\right) D_j^{-1} Z_A = O\left(\frac{1}{nm'}\right) \cdot D_j^{-1} \cdot O_p(\sqrt{n}), \quad (26)$$

where the spectral norm of $D_j^{-1}$ is bounded (i.e., $\|D_j^{-1}\|_{op} = O(1)$).

For $(B)$, define the random variable for the new sample as

$$Z_k = \sum_{k=1}^{\Delta n} x_{i,j+}(r_{k,j}(\beta(\tau))), \quad k \in \tilde{Z}_j^{(k)},$$

where $\tilde{Z}_j^{(k)}$ contains $\Delta n = n_j - n = \frac{n}{m'-1}$ independent samples.

Since the samples are independently and identically distributed, we know that the mean $\mathbb{E}[r_{k,j}(\beta_j^\star(\tau))] = 0$ and the variance $\mathrm{Var}(Z_k) = \Delta n \cdot \mathbb{E}\left[x_{i,j+}x_{i,j+}^\top \cdot \mathrm{Var}(r_j(\beta_j^\star(\tau)) \mid x_{i,j+})\right] = \Delta n \cdot \tau(1-\tau) \cdot \mathbb{E}[x_{i,j+}x_{i,j+}^\top]$. Therefore, by the Central Limit Theorem, we have that

$$\frac{1}{\sqrt{\Delta n}} Z_k \xrightarrow{d} \mathrm{N}(0, \tau(1-\tau) \cdot \mathbb{E}[x_{i,j+}x_{i,j+}^\top]).$$

Therefore, the order of the original sum is:

$$Z_k = O_p(\sqrt{\Delta n}). \tag{27}$$

Bringing equation (27) into (23), we have

$$(B) = \frac{1}{n_j} D_j^{-1} Z_k = O\left(\frac{1}{nm'}\right) \cdot D_j^{-1} \cdot O_p(\sqrt{n}). \tag{28}$$

Combining the orders of $(A)$ and $(B)$, we have

$$\|\tilde{\beta}_j(\tau) - \tilde{\beta}_j^{\mathrm{new}}(\tau)\| \le \|(A)\| + \|(B)\| + o_p(n^{-1/2}) = O_p(n^{-1/2}) + O_p(n^{-1/2}) + o_p(n^{-1/2}),$$

then

$$\|\tilde{\beta}_j(\tau) - \tilde{\beta}_j^{\mathrm{new}}(\tau)\| = O_p(n^{-1/2}).$$

The global perturbation induced by removing agent $m$ propagates through the aggregated estimator as

$$\Delta_j = \widehat{\beta}^{OSW}(\tau) - \widehat{\beta}^{OSW}(\tau)^{\backslash m} = \left(\sum_{m=1}^M T_m^\top W_m T_m\right)^{-1} \sum_{j \in I_m} T_j^\top W_j \left(\tilde{\beta}_j(\tau) - \tilde{\beta}_j^{\mathrm{new}}(\tau)\right).$$

Here $\widehat{\beta}^{OSW}(\tau)^{\backslash m}$ denotes the aggregation parameter obtained by training with all observations after deleting the data of the $m$th agent and reassigning it, as shown in Definition 4.10. Then, combining the spectral paradigm of the inverse of the global aggregation matrix with the summation term, we have

$$\|\Delta_j\| \le \left\|\left(\sum_{m=1}^M T_m^\top W_m T_m\right)^{-1}\right\|_{\mathrm{op}} \cdot \left\|\sum_{j \in I_m} T_j^\top W_j(\tilde{\beta}_j(\tau) - \tilde{\beta}_j^{\mathrm{new}}(\tau))\right\|.$$

Under the spectral norm control $\|W_j\|_{\mathrm{op}} \le C_1$, $\|T_m\|_{\mathrm{op}} \le \sqrt{1 + \|\Sigma_{m+}^{-1}\Sigma_{m\pm}\|_{\mathrm{op}}^2} \le C_2$, then we have $\|\sum_{m=1}^M T_m^\top W_m T_m\|_{\mathrm{op}} \le MC_3$. It follows that

$$\left\|\sum_{j \in I_m} T_j^\top W_j(\tilde{\beta}_j(\tau) - \tilde{\beta}_j^{\mathrm{new}}(\tau))\right\| \le m' \cdot C_1 \cdot C_2 \cdot O_p(n^{-1/2}) = O_p(n^{-1/2}).$$

Then, we derive

$$\|\Delta_m\| \le \frac{1}{MC_3} \cdot \sum_{j \in I_m} \|\tilde{\beta}_j(\tau) - \tilde{\beta}_j^{\mathrm{new}}(\tau)\| = O_p(N^{-1/2}).$$

Leveraging the Lipschitz continuity of the quantile loss $\rho_\tau$, the stability of the global estimator satisfies

$$\left|\rho_\tau(y - x^\top \widehat{\beta}^{OSW}(\tau)) - \rho_\tau(y - x^\top \widehat{\beta}^{OSW}(\tau)^{\backslash m})\right| \le \|x\| \cdot \|\Delta_m\| \le C_4 \cdot O_p(N^{-1/2}),$$

yielding the stability constant bound

$$\mu(m) \le C_4 \cdot O_p(N^{-1/2}) = O_p(N^{-1/2}), \quad \mu = \max_m \mu(m) = O_p(N^{-1/2}).$$

Decompose the generalization error as

$$R(\widehat{\beta}^{OSW}(\tau)) - \widehat{R}(\widehat{\beta}^{OSW}(\tau))$$

$$= \underbrace{R(\widehat{\beta}^{OSW}(\tau)) - \widehat{R}_{\text{dis}}(\widehat{\beta}^{OSW}(\tau))}_{\text{Stability Term}}$$

$$+ \underbrace{\widehat{R}_{\text{dis}}(\widehat{\beta}^{OSW}(\tau)) - E(\widehat{R}_{\text{dis}}(\widehat{\beta}^{OSW}(\tau))) + E(\widehat{R}_{\text{dis}}(\widehat{\beta}^{OSW}(\tau))) - \widehat{R}(\widehat{\beta}^{OSW}(\tau))}_{\text{Statistical Error}}.$$

For stability term, let $D = R(\widehat{\beta}^{OSW}(\tau)) - \widehat{R}_{\text{dis}}(\widehat{\beta}^{OSW}(\tau))$. Here

$$E[D] = E[R(\widehat{\beta}^{OSW}(\tau)) - \widehat{R}_{\text{dis}}(\widehat{\beta}^{OSW}(\tau))]$$

$$= \frac{1}{n} \sum_{i=1}^{n} E[\rho_\tau(y_- (x)^\top \widehat{\beta}^{OSW}(\tau)) - \rho_\tau(y - (x)^\top \widehat{\beta}^{OSW}(\tau)^{\backslash m})] \leq \mu.$$

Applying a block-wise McDiarmid inequality, we obtain

$$\mathbb{P}\left(D - E[D] \geq t\right) \leq \exp\left(-\frac{2t^2}{m \cdot (\mu^2)}\right).$$

Then

$$\mathbb{P}\left(R(\widehat{\beta}^{OSW}(\tau)) - \widehat{R}_{\text{dis}}(\widehat{\beta}^{OSW}(\tau)) \geq \mu + t\right) \leq \exp\left(-\frac{2t^2}{m \cdot (\mu^2)}\right).$$

Let $\delta = \exp\left(-\frac{2t^2}{m \cdot (\mu^2)}\right)$, solving for $t$ yields:

$$t = \sqrt{\frac{m \cdot \mu^2 \cdot \ln(1/\delta)}{2}}.$$

Substituting $\mu = O_p(N^{-1/2})$, we get:

$$t = O\left(\sqrt{\frac{m \cdot N^{-1} \cdot \ln(1/\delta)}{2}}\right) = O\left(\sqrt{\frac{\ln(1/\delta)}{n}}\right).$$

Then, with probability at least $1 - \delta$, we have that

$$R(\widehat{\beta}^{OSW}(\tau)) - \widehat{R}_{\text{dis}}(\widehat{\beta}^{OSW}(\tau)) \leq O_p(n^{-1/2}) + O\left(\sqrt{\frac{\ln(1/\delta)}{n}}\right).$$

For the statistical error term, we define $D_1 = \widehat{R}_{\text{dis}}(\widehat{\beta}^{OSW}(\tau)) - E(\widehat{R}_{\text{dis}}(\widehat{\beta}^{OSW}(\tau)))$ and $D_2 = E(\widehat{R}_{\text{dis}}(\widehat{\beta}^{OSW}(\tau))) - \widehat{R}(\widehat{\beta}^{OSW}(\tau))$.

For $D_2$, according to the definition of stability and the linear nature of expectation, one has

$$D_2 = \mathbb{E}[\widehat{R}_{\text{dis}}(\widehat{\beta}^{OSW}(\tau))] - \widehat{R}(\widehat{\beta}^{OSW}(\tau)) = \frac{1}{N} \sum_{i=1}^{N} \mathbb{E}\left[\ell(\widehat{\beta}^{OSW}(\tau)^{\backslash m}, z) - \ell(\widehat{\beta}^{OSW}(\tau), z)\right].$$

Since the difference between $\widehat{\beta}^{OSW}(\tau)^{\backslash m}$ and $\widehat{\beta}(\tau)$ is guaranteed by parameter estimation consistency ($\|\widehat{\beta}^{OSW}(\tau)^{\backslash m} - \widehat{\beta}^{OSW}(\tau)\| = O_p(n^{-1/2})$), combined with the Lipschitz continuity of the loss function:

$$\mathbb{E}[\ell(\widehat{\beta}^{OSW}(\tau)^{\backslash m}, z) - \ell(\widehat{\beta}^{OSW}(\tau), z)] \leq C_4 \cdot \mathbb{E}[\|\widehat{\beta}^{OSW}(\tau)^{\backslash m} - \widehat{\beta}^{OSW}(\tau)\|] = O(n^{-1/2}).$$

Therefore,

$$D_2 = \mathbb{E}[\widehat{R}_{\mathrm{dis}}(\widehat{\beta}^{OSW}(\tau))] - \widehat{R}(\widehat{\beta}^{OSW}(\tau)) = O(n^{-1/2}).$$

For $D_1$, according to the central limit theorem, we have

$$D_1 = O(n^{-1/2}).$$

Then

$$\widehat{R}_{\mathrm{dis}}(\widehat{\beta}^{OSW}(\tau)) - \widehat{R}(\widehat{\beta}^{OSW}(\tau)) = O_p\left(\frac{1}{\sqrt{n}}\right) = O_p(n^{-1/2}).$$

Combining both terms yields the final convergence rate:

$$R(\widehat{\beta}^{OSW}(\tau)) - \widehat{R}(\widehat{\beta}^{OSW}(\tau)) = O_p(n^{-1/2}) + O_p(n^{-1/2}) = O_p(n^{-1/2}).$$

$\square$

