# OpenReview forum: "One-Shot Weighted Ensemble Estimation for Federated Quantile Regression: Optimal Statistical Guarantees under Heterogeneous Structured Data"
_ICLR.cc/2026/Conference — Submitted to ICLR 2026_

### Official Review · Reviewer_eSZq · 2025-10-16

**Soundness:** 4
**Presentation:** 4
**Contribution:** 4
**Rating:** 8
**Confidence:** 4

**Summary:**

This paper develops a federated quantile regression framework for heterogeneous data and provide corresponding theoretical guarantees.

**Strengths:**

While some assumptions are strong (e.g., Assumption 4.1), the authors provide a comprehensive analysis with detailed theoretical justification.

**Weaknesses:**

I'm wondering whether it is possible to relax Assumption 4.1 to isotropic sub-Gaussian features? If not, what are the restrictions in the proof?

Another question is that, in the proof, how should you handle the o_p terms if M can also asymptotically increase? Based on previous literature, e.g., Volgushev, Stanislav, Shih-Kang Chao, and Guang Cheng. "Distributed inference for quantile regression processes." (2019): 1634-1662. they provide an upper bound on the number of workers. I'm wondering if there is a similar result like that in the federated scenario?

**Questions:**

Please answer my questions in the weakness section.

---

> ### Author Response · Authors · 2025-11-21
>
> Dear Reviewer,
>
> Thank you for your thorough review and constructive feedback. We appreciate your insights, and we have addressed your concerns and questions in our response below:
>
> # **Response to weakness 1**
>
> We thank the reviewer for the insightful comments. Indeed, it is possible to extend the established theoretical results from the Gaussian setting to a sub-Gaussian setting. However, doing so requires substantially more sophisticated tools, such as linear projection techniques and matrix concentration inequalities. For example, in the Gaussian case, we have
> $x_{i, m-}=\Sigma_{i \pm}^{\top} \Sigma_{i+}^{-1} x_{i, m+}+\mathbf{v}, \quad \mathbf{v} \sim N\left(0, \Gamma_{i-}\right),$
> where $\Gamma_{i-}=\Sigma_{i-}-\Sigma_{i \pm}^{\top} \Sigma_{i+}^{-1} \Sigma_{i \pm}$ is the Schur complement. This exact decomposition does not hold under sub-Gaussian designs; linear projection techniques may help to overcome this issue. We empirically evaluate the performance of the proposed method under a sub-Gaussian setting in Appendix A.5 of the revised version. The results demonstrate both the potential for extending our framework to this setting and the good performance of our method relative to other methods.
>
>
> Although extending the results to sub-Gaussian settings is both novel and important, the associated technical challenges are nontrivial and fall beyond the intended scope of this work. We therefore leave this work for the future.  To the best of our knowledge, our work is the first to investigate federated quantile regression with such heterogeneous structured features. Therefore, as a starting point, we impose Gaussian design assumptions to keep the setting analytically tractable. We have added a discussion of these points in the revised version of Section 6, which is marked in blue.
>
>
> # **Response to weakness 2**
>
> These two questions pertain to the non-asymptotic analysis of the learned global parameters, which can be addressed by studying the sample complexity under more detailed settings and derivations. Conducting such an analysis involves different technical tools and falls outside the scope of the current paper.

---

> > ### Author Response · Authors · 2025-11-28
> >
> > We would like to sincerely thank the reviewer once again for recognizing the novelty of our work and for thoughtfully pointing out the limitations of the current paper, particularly the possibility of relaxing the Gaussian assumptions on the features. This highlights the nontrivial extension of our work. We hope that our detailed responses have fully addressed the reviewer’s concerns. If the reviewer has any further questions, we would be more than happy to provide additional clarification.

---

### Official Review · Reviewer_h1sK · 2025-10-19

**Soundness:** 2
**Presentation:** 3
**Contribution:** 3
**Rating:** 6
**Confidence:** 2

**Summary:**

This paper proposes a novel one-shot weighted ensemble estimator for FQR that effectively handles heterogeneously structured data. By incorporating an optimal weighting scheme, the method achieves communication efficiency while establishing optimal statistical guarantees, which is validated through extensive experiments on both synthetic and real-world data.

**Strengths:**

This paper introduces a communication-efficient FQR method, specifically designed for a unique form of heterogeneous structured data. Its primary strength lies in its theoretical results, which demonstrate that the proposed estimator achieves both asymptotic and non-asymptotic theoretical guarantees.

**Weaknesses:**

There are some critical issues the authors need to address, including: (1) strong modeling assumptions and an unusual definition of heterogeneity; (2) questionable stability arguments with respect to non-i.i.d. agents; and (3) a simple experimental design with limited experimental baselines.

**Questions:**

1. The assumption of a linear data-generation process in the problem setup is overly strong and does not align well with most real-world scenarios.
2. The definition of heterogeneity employed in the paper is somewhat unusual. In typical federated learning settings, it is more common to assume heterogeneity in the model parameter (e.g., $\beta^*$).
3. The paper is mathematically dense and uses a large number of notations. It would be beneficial to include a dedicated section that introduces and summarizes all of the notations used throughout the manuscript.
4. In Remark 3.1, the authors present the communication cost of the proposed method, but they do so without providing comparison for other FQR methods. For readers less familiar with this field, it would be helpful if the authors compared the communication overhead of their method with that of existing methods, and discussed whether the communication cost becomes especially large in applications with very large problem scales.
5. In Section 4.2, we know that traditional sample-level algorithmic stability typically assumes each sample is i.i.d. However, in the present setting the authors consider agent-level stability where each agent may be non-i.i.d. Therefore, it is unclear whether the standard algorithmic stability framework can be directly applied to this setup.
6. In the numerical experiments, the authors compare the proposed method only against two very simple baselines (“Naive-local” and “Naive-OSFL”). The limited number and simplicity of these baselines weaken the evidence for the superiority of the proposed method. Additionally, the authors are encouraged to report mean values together with standard errors (over multiple runs) in order to demonstrate the robustness of their results.

---

> ### Author Response · Authors · 2025-11-21
>
> Dear Reviewer,
>
> Thank you for your thorough review and constructive feedback. We appreciate your insights, and we have addressed your concerns and questions in our response below:
>
> # **Response to question 1**
>
> We thank the reviewers for their insightful comments regarding the linear data-generating assumption. While real-world scenarios can be nonlinear, the linear assumption remains a widely used and essential benchmark, providing a tractable foundation for theoretical analysis and capturing a broad class of practical problems. Moreover, many methods for nonlinear modelling (e.g., generalized additive models and shallow neural networks) inherently rely on linear components or local linear approximations. Thus, we focus on the linear setting in this work and leave extensions to nonlinear models for future study.
>
> # **Response to question 2**
>
> Thank you for raising this point. We agree that the heterogeneity considered in our paper differs from the classical setting, where model parameters vary across agents. Our focus is motivated by practical constraints, where it is often infeasible for all agents to access or collect the same set of features due to implementation challenges, policy restrictions, financial limitations, or differences in domain expertise. Consequently, agents may only observe subsets of the data, leading to diverse feature sets. We highlight that this type of heterogeneity offers a new perspective on federated learning, motivating the need for an aggregation estimator that can effectively leverage distinct feature sets while achieving both local and global optimality.
>
>
> # **Response to question 3**
>
> Thank you for pointing this out. We have clarified the notation as suggested in the revised version and included a notation table as below to define each symbol used and its meaning, to improve overall readability.
>
> **Table 1: Notations and their meaning**
>
> | Notations              | Meaning                                                         |
> |------------------------|-----------------------------------------------------------------|
> | τ                      | quantile level                                                  |
> |  $y _ {i,m}$           | i-th observed response for agent $ m $                       |
> | $ x _ {i,m+}, x_{i,m-} $ | local observed and unobserved features for agent $ m $       |
> | $ \beta ^ *(\tau) $    | true parameters                                                 |
> | $ \tilde{\beta} _ m(\tau) $ | local QR estimator for agent $ m $                       |
> | $ \hat{\beta}(\tau; \Omega(W)) $| global estimator                                   |
> | $N, n $            | total and local sample size                                     |
> | $M $                | number of agents                                                |
> | $ \Sigma _ m ^ + $       | observed covariance for agent $ m $                          |
> | $ \Pi _ m $           | permutation matrix for agent $ m $                            |
> | $ \Pi _ {m+}, \Pi _ {m-} $ | extract observed and unobserved features (covariates) for agent $ m $  |
>
>
> # **Response to question 4**
>
> Thank you for the suggestion. We would like to clarify that the existing multi-round communication methods are not applicable to the type of heterogeneous structured data considered in this paper. To ensure a fair and meaningful comparison, we therefore focus on comparing our algorithm with one-shot methods discussed in the paper. In the revised manuscript, we have added a table as shown below in Section 5 to illustrate the communication costs associated with each algorithm more clearly.
>
> **Table 2: Computation cost for agent $m$ under different methods**
>
> | Methods      | Communication cost |
> |--------------|--------------------|
> | Naive-Local  |     --                 |
> | Naive-OSFL   |     $O(d_i)$           |
> | UW-OSW       |    $O(d_i^2)$         |
> | Centralized  |    --                 |
> | DA-OSW       |     $O(d_i^2)$         |
> | OSW          |     $O(d_i^2)$         |

---

> > ### Author Response · Authors · 2025-11-21
> >
> > # **Response to question 5**
> >
> > Thank you for pointing this out. We confirm that the standard algorithmic stability framework continues to apply in our setting.
> >
> > In the classical setting, sample-level uniform stability measures how much the output of a learning algorithm changes when a single training sample is replaced by another i.i.d. sample. In our federated setting, however, the natural perturbation unit is not an individual sample but an entire subset (or block) of samples belonging to an agent. Accordingly, our stability notion evaluates how the global estimator changes when the full local dataset of one agent is perturbed. The deviation is therefore taken with respect to a set of samples rather than a single sample, but crucially, each agent’s local dataset is still an i.i.d. sample from the same underlying distribution. Thus, the i.i.d. assumption underlying classical stability arguments remains valid.
> >
> > Another way to interpret the agent-level stability is through the structure of the one-shot estimator. In our setting, the global estimator is constructed by aggregating local estimators produced independently by each agent. It is more natural to study the sensitivity of the global estimator to perturbations of entire local estimators rather than individual samples. This motivates an agent-level stability notion. Because each local estimate is computed from an i.i.d. set of samples, the standard algorithmic stability framework continues to apply.
> >
> >
> > # **Response to question 6**
> >
> > We highlight that the proposed method can simultaneously address the following three aspects: (1) heterogeneous feature structures across agents, (2) leverage cross-agent correlations for collaborative model training, and (3) communication constraints inherent in federated settings. To the best of our knowledge, existing works rarely address all three aspects at once.
> >
> > In the revised experimental setup, we mitigate the design limitations as follows: (1) UA-OSW: use a single machine to concentrate all data without missing features, providing an optimal baseline across algorithms; (2) UW-OSW: within the algorithm framework, replace each $W_m$ with diagonal elements that are 1 and the remaining elements that follow a standard normal distribution to verify the optimality of $W_m$; (3) DA-OSW: examine the role of $T_m$ by substituting it with $I_d$. Additionally, we have added standard errors to demonstrate the robustness of the proposed method in Section 5. All the newly added results (Section 5 and Appendix A) are marked in blue in the revised paper. Please refer to the updated version.

---

> > > ### Author Response · Authors · 2025-11-28
> > >
> > > We sincerely appreciate the time and effort the reviewer has devoted to reviewing our paper. The suggestion to include a notation table and additional numerical results has greatly improved the clarity and quality of the manuscript. We hope that the revisions and detailed responses adequately address the reviewer’s concerns. If any further questions or clarifications are needed, we would be more than happy to provide additional explanations.

---

### Official Review · Reviewer_QhDW · 2025-10-20

**Soundness:** 2
**Presentation:** 2
**Contribution:** 2
**Rating:** 2
**Confidence:** 3

**Summary:**

The authors propose a quantile regression framework for federated learning with structured missing data and provide a one-shot weighted ensemble estimation algorithm. Theoretical guarantees, including convergence rates and asymptotic normality, are presented and are noted as being solid and comprehensive.

**Strengths:**

*   Well-established theoretical results with detailed proofs covering both convergence rates and asymptotic normality.
*   The proposed algorithm addresses the challenge of structured missing data in a federated learning context.

**Weaknesses:**

*   **Lack of Clear Innovation:** The core methodological innovation appears limited, as the algorithm seems to be a direct adaptation of existing work (Chen Cheng, 2023) by merely substituting the gradient and Hessian calculations for the quantile loss, without a significant abstraction or generalization.
*   **Insufficient Motivation for Quantile Regression:** The paper lacks a dedicated discussion on the specific properties and challenges of the quantile loss function (e.g., non-differentiability, computational aspects), failing to justify why this specific loss is used and how its peculiarities are handled.
*   **Inadequate Experimental Evaluation:** The experiments are considered thin. Key baselines and analyses are missing, including:
    *   Comparison with a pooled estimator.
    *   Comparison with an estimator using fully observed data.
    *   Experimental validation of the theoretical asymptotic normality.
    *   Experiments on heavy-tailed distributions (e.g., Cauchy).
    *   Analysis of estimation error variability.
    *   Sensitivity analysis regarding the number of quantiles (M).

**Questions:**

1.  What is the fundamental methodological advancement beyond the work of Chen Cheng (2023)? Specifically, how does the proposed framework abstractly generalize the problem for a broader class of loss functions, or, if it is specific to quantile regression, what unique technical challenges does it solve?
2.  Why is there no discussion on the inherent properties of the quantile loss function (like non-differentiability), and how does the proposed algorithm effectively manage these challenges?
3.  Can the authors provide more comprehensive experiments, including the missing baselines (full-data ) and a verification of the asymptotic normality results?
4.  How does the method perform under heavy-tailed noise distributions, and how does the number of quantiles (M) impact the estimation stability and accuracy?

---

> ### Author Response · Authors · 2025-11-21
>
> Dear Reviewer,
>
> Thank you very much for your thoughtful review and constructive feedback. The weaknesses and questions you highlighted can be summarized as follows. We truly appreciate your insights, and we have carefully addressed your questions and concerns in our response below:
>
> # **1. Motivation for considering quantile regression**
>
> Quantile regression is meaningful even in standard (non-federated) learning scenarios. It provides a powerful modelling framework for estimating conditional quantiles, offering a more comprehensive understanding of response distributions than classical mean regression. Mean regression captures only average behaviour, quantile regression characterizes how predictors influence different points of the outcome distribution. This is particularly important in federated learning setting, where applications often involve tail behaviour or extreme events, such as hydrology, risk assessment, and medical diagnostics. This makes quantile-based inference more relevant than mean regression.
>
> Due to space constraints and the well-recognized significance of quantile regression in the existing machine learning and statistics literature, we provided only a brief discussion of its motivation in the revised version, marking them in blue (page 1).
>
> # **2. Comparison with Chen Cheng (2023)**
>
>  Both Cheng’s work and ours investigate federated learning with heterogeneous structured features. However, whereas Cheng focuses on federated mean regression (quadratic loss), our work addresses federated quantile regression (quantile loss), which introduces substantially different technical challenges.
>
> Theoretical challenges.
> Under mean regression, the local estimator admits a closed-form expression. Cheng’s analysis, for example, characterizing the asymptotic covariance of the learned estimators and deriving the optimal weight matrix, relies heavily on this closed-form structure. In contrast, the quantile regression estimator lacks a closed-form representation, and the nonsmooth nature of the quantile loss further complicates the analysis. As a result, entirely new techniques are required to derive the covariance structure of the collaborative estimator and determine optimal aggregation weights. In particular, we make essential use of advanced tools such as the Bahadur linear representation of local quantile estimators to overcome these challenges.
>
> Algorithmic challenges.
> Because no closed form exists for the local quantile estimator, we must solve a linear programming problem in Step 3 of Algorithm 1; in contrast, Cheng's method computes the local estimator directly from the closed-form expression. Moreover, the quantile loss complicates the computation of the optimal weight matrix $W^*$ in Step 8, which requires estimating the conditional density $f_{\xi_{i, m}}\left(0 \mid x_{i, m+}\right)$. This is achieved using a one-dimensional kernel density estimator based on local residuals.
>
> Additionally, while the core ideas of our framework can in principle be extended to other loss functions, doing so would introduce new technical and tractability challenges. For example, establishing the asymptotic normality of the learned global estimators or deriving the corresponding optimal weight matrix $W^\star$ under a general loss function would require substantially different analytical tools. These extensions are nontrivial, as they depend on the specific smoothness, and noise properties associated with each loss.

---

> > ### Author Response · Authors · 2025-11-21
> >
> > # **3. Experimental Evaluation**
> >
> > We thank the reviewers for their helpful suggestions regarding additional experiments. In the revised version, we have strengthened the empirical evaluation in multiple ways. First, we used a single machine to concentrate all data without missing features, providing an optimal baseline across algorithms (Centralized). We also added the prediction variability. Second, within the algorithm framework, replace each $W_m$ with diagonal elements that are 1 and the remaining elements that follow a standard normal distribution to verify the optimality of $W_m$ (UW-OSW), and similarly, examine the role of $T_m$ by substituting it with $I_d$ (DA-OSW). These comparisons help clarify the effectiveness of the proposed weighting and transformation matrix.
> >
> > **Asymptotic normality：**  As demonstrated in Figure 2 of the manuscript, we have verified asymptotic normality by showing that the empirical mean and variance of the estimators converge to their theoretical values as the sample size increases.
> >
> > **Heavy-tailed distributions.：** The manuscript already includes results under heavy-tailed noise, $t$-distribution. Due to space limitations, these results are showed in Appendix A.4. In the revised version, we have highlighted the corresponding text in blue for easier identification. In addition, as the reviewer suggested, we also added the numerical results under the Cauchy distribution. These results are presented in Section 5 of the revised paper.
> >
> > **Sensitivity to the number of quantiles (M)：**  We would like to clarify that the quantile level corresponds to $\tau$, whereas $M$ denotes the number of agents. The manuscript already evaluates performance across three quantile levels $\{0.2,0.5,0.8\}$ under different settings of the noise distribution (Section 5).
> >
> >  All the newly added results (Section 5 and Appendix A) are marked in blue in the revised paper.

---

> > > ### Comment · Reviewer_QhDW · 2025-11-24
> > >
> > > I appreciate the authors' response, which has addressed some of my concerns to a certain extent and further improved the paper. I also hope the authors can emphasize the characteristics of the quantile loss function—such as the absence of an explicit solution—and highlight their algorithmic innovations in overcoming these features, so as to underscore the differences from the method proposed by Chen Cheng (2023).
> > >
> > > Additionally, a minor suggestion: some of the figures, such as Figures 2, 5, 7, 9, and 11, could be further refined and beautified to make them more concise and save space.
> > >
> > > I will increase my score to 4.

---

> > > > ### Author Response · Authors · 2025-11-25
> > > >
> > > > We are pleased to hear that our previous response has addressed the reviewer’s concerns. Following the additional suggestions, we have made two updates in the revised paper, which are marked in blue:
> > > >
> > > >  **Underscore the differences:**
> > > >
> > > > We added the following paragraph at the end of Section 3 to emphasize the characteristics of the quantile loss function and to differentiate our algorithm from those in Cheng (2023):
> > > >
> > > >    Compared with Cheng's work, which considers a quadratic loss for each local agent, the quantile loss used in our framework introduces substantial computational challenges in Algorithm 1. Because no closed-form expression exists for the local quantile estimator, Step 3 requires solving a linear programming problem, whereas Cheng's estimator can be computed directly via a closed-form solution. Furthermore, obtaining an estimate of the optimal weight matrix $W ^ *$ in Step 8 requires estimating the conditional density $f _ {\xi _ {i, m}}\left(0 \mid x _ {i, m+}\right)$, a step unnecessary in Cheng's framework.
> > > >
> > > > **Figure refinement:**
> > > >
> > > > We refined Figures 2, 5, 7, 9, and 11 to save more space.
> > > >
> > > > We sincerely thank the reviewer again for the constructive and thoughtful feedback, which has helped improve both the clarity and quality of the paper. Please let us know if there are any remaining questions or further suggestions, we would be more than happy to address them.

---

> > > > > ### Comment · Reviewer_QhDW · 2025-11-26
> > > > >
> > > > > Thank you for your detailed responses and for incorporating the revisions into the manuscript.
> > > > > However, a new question has come to mind regarding Figure 2.
> > > > > Why does the mean error at the 0.2 quantile appear to be lower than those at the 0.5 and 0.8 quantiles? This seems somewhat counterintuitive from a quantile regression perspective, as one would generally expect the estimation error to exhibit a monotonic relationship with the weighting term $1/\tau(1-\tau)$. A similar pattern can also be observed in Figure 5. Could you please clarify this behavior?

---

> > > > > > ### Author Response · Authors · 2025-11-27
> > > > > >
> > > > > > We thank the reviewer for the thoughtful comments. We would first like to clarify that the left panel of Figure 2 reports the estimation error of the mean, i.e., $\||\textrm{E}(\widehat{\beta}(\tau; \Omega(\widehat{W})))-\beta ^ {\star}(\tau)\||_ 1$, while the right panel reports the estimation error of the covariance structure, i.e., $\||\textrm{cov}(\sqrt{n}(\widehat{\beta}(\tau; \Omega(\widehat{W}))-\beta ^ {\star}(\tau)) - C(\Omega(\widehat{W}))\|| _ F$.
> > > > > > Regarding the reviewer’s concern about monotonicity across quantile levels, such behavior should not be expected for either quantity. The mean estimation error generally does not exhibit a monotone trend in $\tau$, as it depends on both the data-generating mechanism and the finite-sample behavior of
> > > > > > $\widehat{\beta}(\tau; \Omega(\widehat{W}))$.
> > > > > >
> > > > > > Similarly, a monotone pattern is not expected for the covariance estimation error. This error also depends on the conditional density of the residuals evaluated at $0$, which itself may also vary with the quantile level $\tau$. In our numerical experiments, this density is estimated by using a one-dimensional kernel density estimator based on the residuals
> > > > > > $r _ {i, m}=y _ {i, m}-x _ {i, m+} ^ {\top} \tilde{\beta} _ m(\tau)$:
> > > > > >  $\widehat{f} _ {\xi _ i, m}\left(0 \mid x _ {i, m+}\right)=\frac{1}{n h _ m} \sum_{i=1} ^ {n} K\left(r _ {i, m} / h _ m\right), $
> > > > > > where $K(\cdot)$ is the Gaussian kernel. As is well known, density estimation becomes less stable for extreme quantiles due to fewer effective observations near the tail regions. Consequently, the covariance estimation error may increase in the tails, preventing any monotone trend across $\tau$.
> > > > > >
> > > > > > we sincerely thank the reviewer again for the constructive and thoughtful feedback, which has helped improve both the clarity and quality of the paper. Please let us know if there are any remaining questions or further suggestions, we would be more than happy to address them.

---

### Official Review · Reviewer_MF5F · 2025-11-04

**Soundness:** 3
**Presentation:** 3
**Contribution:** 3
**Rating:** 4
**Confidence:** 2

**Summary:**

The paper proposes a one-shot Federated Quantile Regression (FQR) method for heterogeneous structured data where agents observe distinct feature subsets. Each agent fits local linear QR on its features  then the server solves a weighted ERM mixing local estimators. Theory claims provided include asymptotic normality for any positive-definite weights; existence of variance-optimal W⋆,  and generalization bound via agent-dependent stability among few more things.

**Strengths:**

1) Relevant problem and one-shot design: FQR with heterogeneous feature access is under-explored;  transformation reconciles partial features via cross-correlations rather than truncation; avoids iterative rounds
2)  Good theory contributions ie: asymptotic normality for broad weights, variance-optimal W⋆ characterization, practical plug-in construction, and agent-perturbation stability bound
3) Transparent communication: Per-agent payload d²ᵢ+3dᵢ explicit which is a good sign  and n-independent

**Weaknesses:**

1) Since this is mainly a theory based paper it seems the major results require gaussian designs and structural coverage requiring that the union of local supports spans the full feature space. Computing the key quantities Tₘ and W⋆ requires access to matrices A and B that depend on unobserved counterfactual data x⁻. Broadly speaking a little gap seems to exist in the theory and empirical part of the paper. More on this in my next point

2) The empirical part of the paper seems a bit thin when compared to the strength of the theory claim: For instance, the paper compares only against weak baselines (Naive-Local and simple averaging), using a single small real-world dataset (California Housing, median quantile τ=0.5). I would appreciate if the experimental section can be expanded to consider more variant setting

**Questions:**

Weaknesses and Questions merged

---

> ### Author Response · Authors · 2025-11-21
>
> Dear Reviewer,
>
> Thank you for your thorough review and constructive feedback. We appreciate your insights, and we have addressed your concerns and questions in our response below:
>
> # **Response to weakness 1**
>
> **1. Gaussian designs:** We agree with the reviewer’s point that the current theory is limited by the Gaussian assumption. We emphasize, however, that establishing the theoretical guarantee, e.g., asymptotic normality of the learned parameters, and the determination of optimal weight matrix $W^\star$ remains technically challenging even under Gaussian features: unlike least squares, the local quantile regression estimator does not admit a closed-form expression, and the non-smoothness of the quantile loss further complicates the analysis. Our results, therefore, require new techniques, such as Bahadur linear representation, beyond those used for federated mean regression. To the best of our knowledge, our work is the first to investigate federated quantile regression with such heterogeneous structured features. Therefore, as a starting point, we impose Gaussian design assumptions to keep the setting analytically tractable. We have added this discussion to the revised version of Section 6, which is marked in blue.
>
> **2. Structural coverage requires that the union of local supports spans the full feature space:** The structural coverage assumption concerns the covariance matrix $\Sigma$ and is, in fact, a very mild requirement. This condition is fundamental in settings with a missing scheme: if a variable is never collected or observed by any agent, then the data contain no information about that feature, and its covariance structure cannot be inferred. Therefore, the assumption does not impose any additional practical constraints; rather, it ensures that the information in $\Sigma$ is recoverable from the available data.
>
> **3. Computing the key quantities $T_m$ and $W^\star$ requires access to matrices $A$ and $B$ that depend on unobserved counterfactual data $x_{-}$:**  The key quantities $T_m$ and $W^\star$ are estimated by aggregating information from all agents in practice. This is justified by the fact that all agents share a common underlying distribution, although $x_{i, m-}$ is unobserved by agent $m$, its contribution can be effectively inferred from peer agents when computing the matrices $A$ and $B$. Additionally, the density $f_{\xi_{i, m}}\left(0 \mid x_{i, m+}\right)$ is estimated using a one-dimensional kernel density estimator based on the residuals $r _ {i, m}=y _ {i, m}-x_ {i, m+} ^  {\top} \tilde{\beta} _ m(\tau)$. Specifically,
> $$ \widehat{f} _ {\xi _ , m}\left(0 \mid x _ {i, m+}\right)=\frac{1}{n h _ m} \sum _ {i=1} ^{n} K\left(r _ {i, m} / h _ m\right), $$
>
> where $K(\cdot)$ is the Gaussian kernel, $K(u)=(2 \pi)^{-1 / 2} \exp \left(-u^2 / 2\right)$, which is
> used throughout the numerical experiments. For the bandwidth $h_m$, we adopt Silverman's rule of thumb
>
> $h_m=1.06 \hat{\sigma}_{r, m} n_m^{-1 / 5}$,
>
> where $\hat{\sigma} _ {r, m}$ is the sample standard deviation of $[r _ {i.m}]_ {i=1}^{n}$, and $n$ is the sample size for agent $m$. Due to space limitations, we added these details in the revised version in Appendix A, which is marked in blue.
>
> # **Response to weakness 2**
>
> We highlight that the proposed method can simultaneously address the following three aspects: (1) heterogeneous feature structures across agents, (2) leverage cross-agent correlations for collaborative model training, and (3) communication constraints inherent in federated settings. To the best of our knowledge, existing works rarely address all three aspects at once.
>
> In the revised experimental setup, we mitigate the design limitations as follows: (1)Centralized: Use a single machine to concentrate all data without missing features, providing an optimal baseline across algorithms; (2)UW-OSW: within the algorithm framework, replace each $W_m$ with diagonal elements that are 1 and the remaining elements that follow a standard normal distribution to verify the optimality of $W_m$; (3) DA-OSW: examine the role of $T_m$ by substituting it with $I_d$.
>
> In addition, we have added experiments with noise following a Cauchy distribution to further demonstrate the effectiveness of our algorithm for heavy-tailed distributions. For real data analysis, we have also added the results across different quantile levels and compared our methods with others to more comprehensively illustrate the performance of the proposed method.
>
> All the newly added results (Section 5, Appendix A.1 - A.4 and Appendix A.6) are marked in blue in the revised paper.

---

> > ### Author Response · Authors · 2025-11-28
> >
> > We would like to sincerely thank the reviewer once again for the constructive feedback and thoughtful comments, especially the suggestion to include additional numerical results to further validate the performance of the proposed method. These suggestions indeed strengthen our paper. We hope that our revisions and detailed responses adequately address the reviewer’s concerns. If there are any remaining questions or clarifications needed, we would be more than happy to address them.

---

### Author Response · Authors · 2025-11-29

Dear Area Chair,

We would like to sincerely thank you and all the reviewers for the time and effort invested in reviewing our paper. We especially appreciate **Reviewer QhDW’s** active engagement during the discussion phase and are pleased that our responses have addressed their concerns, leading to an updated score from 2 to 4.

The main concerns raised by the reviewers are: (1) the Gaussian assumptions on the features and the possibility of extending our results to sub-Gaussian settings; (2) the need for additional baseline comparisons in numerical experiments.

(1) We emphasize that, even under Gaussian features, establishing the theoretical guarantee, such as asymptotic normality of the learned parameters and the determination of optimal weight matrix $W^\star$ remains technically challenging compared to the quadratic loss. As a starting point, we impose Gaussian assumptions to keep the setting analytically tractable and provide a thorough theoretical analysis. Additional details are discussed in our response to **Reviewer MF5F**. As noted in our response to **Reviewer eSZq**, extending our results to sub-Gaussian settings is an important and nontrivial direction, which requires more advanced techniques, such as linear projection methods and matrix concentration inequalities. Furthermore, we have explored the relaxation of Gaussianity through additional experiments and theoretical discussion in Section A.5.

(2) To strengthen the empirical evaluation, we incorporate several additional baselines: Centralized (using full-feature centralized data as an optimal benchmark), UW-OSW (replacing each $W_m$’s diagonal with ones and filling off-diagonal elements with Gaussian noise to assess weight optimality), and DA-OSW (replacing $T_m$ with $I_d$ to examine its role). We also included standard errors in Section 5 and added more analysis in real-world experiments to further demonstrate the robustness and superiority of our method.

All the newly added experimental results and content are marked in blue in the revised paper.

Thank you all again for the time and dedication in the review process.

---

### Meta-Review · Area_Chair_f8tV · 2026-01-04

**Summary:**

This paper studies the federated quantile regression problem for heterogeneous structured data, where agents observe distinct feature subsets. Both theoretical analysis and empirical evaluation are provided to demonstrate the performance of the proposed algorithm.

The key concerns include an overly simple evaluation, strong assumptions, and an uncommon problem setup. For example, the proposed algorithm is evaluated on only one real-world dataset with a small scale. Moreover, the theoretical analysis relies on a Gaussian assumption, which limits its applicability to real-world settings. In addition, the assumption that agents observe distinct feature subsets is not common in federated learning.

After the rebuttal, these key concerns remain unresolved. For example, the authors did not provide additional results on large-scale real-world datasets. The concerns regarding the assumptions and problem setup also remain, despite the authors’ additional explanations.

Furthermore, from the area chair’s perspective, the theoretical results are not sufficiently strong to warrant acceptance. In particular, given that the focus of the paper is federated learning, the theoretical analysis should reflect how the federated learning setting affects the performance of the algorithm. For instance, it should explicitly address how the number of workers and data heterogeneity influence the theoretical guarantees. Otherwise, the analysis does not substantially differ from that of a non-federated learning setting.

Due to these issues, I recommend rejection.

**Reviewer Concerns:**

The key concerns include an overly simple evaluation, strong assumptions, and an uncommon problem setup. For example, the proposed algorithm is evaluated on only one real-world dataset with a small scale. Moreover, the theoretical analysis relies on a Gaussian assumption, which limits its applicability to real-world settings. In addition, the assumption that agents observe distinct feature subsets is not common in federated learning.

After the rebuttal, these key concerns remain unresolved. For example, the authors did not provide additional results on large-scale real-world datasets. The concerns regarding the assumptions and problem setup also remain, despite the authors’ additional explanations.

**Reviewer Scores:**

Reviewer QhDW claimed to raise the score from 2 to 4.

---

### Decision · Program_Chairs · 2026-01-26

Reject